# Interpretable factorization of clinical questionnaires to identify latent factors of psychopathology

**Ka Chun Lam**                                            *kachun.lam@nih.gov*
*Machine Learning Core*
*National Institute of Mental Health, National Institutes of Health*

**Francisco Pereira**                                      *francisco.pereira@nih.gov*
*Machine Learning Core*
*National Institute of Mental Health, National Institutes of Health*

**Bridget W Mahony**                                       *bridgetwmahony@gmail.com*
*Section on Developmental Neurogenomics, Human Genetics Branch*
*National Institute of Mental Health, National Institutes of Health*

**Armin Raznahan**                                         *raznahana@mail.nih.gov*
*Section on Developmental Neurogenomics, Human Genetics Branch*
*National Institute of Mental Health, National Institutes of Health*

**Reviewed on OpenReview:** *https://openreview.net/forum?id=1Yq6INJwiO*

## Abstract

Psychiatry research seeks to understand the manifestations of psychopathology in behavior, as measured in questionnaire data, by identifying a small number of latent factors that explain them. While factor analysis is the canonical tool for this purpose, the resulting factors may not be interpretable, and may also be subject to confounding variables. Moreover, missing data are common, and explicit imputation is often required. To overcome these limitations, we introduce Interpretability Constrained Questionnaire Factorization (ICQF), a non-negative matrix factorization method with regularization tailored for questionnaire data. Our method aims to promote factor interpretability and solution stability. We provide an optimization procedure with theoretical convergence guarantees, and an automated procedure to determine latent dimensionality accurately. We validate these procedures using realistic synthetic data. We demonstrate the effectiveness of our method in a widely used general-purpose questionnaire, in two independent datasets (the Healthy Brain Network and Adolescent Brain Cognitive Development studies). Specifically, we show that ICQF preserves diagnostic information across a range of disorders, outperforming competing methods for smaller dataset sizes, and improves interpretability, as assessed by our clinical research collaborators and co-authors. This suggests that the regularization in our method matches domain characteristics, in addition to satisfying qualitative desiderata.

## 1 Introduction

Standardized questionnaires are a common tool in psychiatric practice and research, for purposes ranging from screening to diagnosis or quantification of severity. A typical questionnaire comprises questions – usually referred to as *items* – reflecting the degree to which particular symptoms or behavioural issues are present in study participants. The items provide evidence for the presence of *latent constructs* giving rise to

the psychiatric problems observed. For many common disorders, there is a practical consensus on constructs. If so, a questionnaire may be organized so that subsets of the items can be added up to yield a *subscale score* quantifying the presence of their respective construct. Otherwise, the goal may be to discover new constructs, by factorizing a questionnaire containing items of interest in a particular domain.

The *factor analysis* (FA) of a questionnaire matrix (#participants × #items) expresses it as the product of a factor matrix (#participants × #factors) and a loading matrix (#factors × #items). The method assumes that answers to items should be correlated, and can therefore be explained in terms of a smaller number of factors. The method yields two real-valued matrices, with uncorrelated columns in the factor matrix. The number of factors is specified a priori, or estimated from data. The values of the factors for each participant can then be viewed as a succinct representation of them.

Interpreting what construct a factor may represent is done by considering its loadings across items. Ideally, if very few items have a non-zero loading, or each item only has a high loading on a single factor, it will be easy to associate the factor with them. The FA solution is often subjected to rotation to accomplish this. In practice, the loadings could be an arbitrary linear combination of items, with positive and negative weights. Factors are real-valued, and neither their magnitude nor their sign are intrinsically meaningful. Beyond this, any missing data will have to be imputed, or the respective items omitted, before FA can be used. Finally, patterns in answers that are driven by other characteristics of participants (e.g. age or sex) are absorbed into factors themselves, acting as confounders, instead of being represented separately or controlled for.

In this paper, we propose to address all of the issues above with a novel matrix factorization method specifically designed for use with questionnaire data, through the following contributions:

**1. Interpretability-Constrained Questionnaire Factorization (ICQF)**    Our method incorporates key characteristics which enhance the interpretability of resulting factors, as conveyed by clinical psychiatry collaborators. These characteristics are translated into mathematical constraints as follows:

- Factor values are within the range of $[0, 1]$, representing the degree of presence of the factor.

- Factor loadings are bounded within the same value range as the original questionnaire responses, facilitating interpretation as answer patterns associated with the factor, rather than arbitrary values.

- The reconstructed matrix adheres to the known value range or observed maximum of the original questionnaire, preventing any entry from exceeding these limits.

- The method directly handles missing data without requiring imputation. Additionally, it allows for the inclusion of pre-specified factors to capture answer patterns correlated with known variables.

**2. Theoretical foundations of ICQF**    ICQF unifies established tools into a single framework that turns clinical interpretability desiderata into factorization constraints. These constraints, on both the factors and the reconstructed matrix, pose algorithmic challenges, which we address with an optimization procedure for ICQF based on alternating minimization with ADMM. Our technical contributions are: (i) a sufficient condition on the ADMM penalty parameter ($\rho \geq \sqrt{2}$) guaranteeing convergence to a KKT point of the constrained formulation; (ii) that, when the latent dimension is correctly specified, the factorization is close to a global optimum with high probability, which we verify on realistic synthetic data; and (iii) that these guarantees extend to the box constraints that promote interpretability. We additionally implement blockwise cross-validation (BCV) to select the number of factors, and show both that a correct count yields a near-global solution and that BCV detects it more precisely than competing methods.

**3. Method evaluation**    We conduct a comprehensive evaluation of ICQF in comparison with state-of-the-art methods on CBCL, a widely used questionnaire to assess behavioral and emotional problems, collected in two independent clinical studies (*HBN* and *ABCD*). We demonstrate the effectiveness of our method on quantitative metrics that reflect preservation of diagnostic information in latent factors, and stability of factor loadings in limited sample sizes or across datasets.

**4. Light-weighted implementation**    We provide a Python implementation of ICQF that can efficiently handle questionnaire datasets in psychology or psychiatry research contexts, with large-scale participant cohorts. The standalone package is available at https://github.com/jefferykclam/ICQF.

## 2 Related Work and Technical Motivation for our Method

The extraction of latent variables (a.k.a. factors) from matrix data is often done through low rank matrix factorizations, such as singular value decomposition (SVD), principal component analysis (PCA) and exploratory Factor Analysis (FA) Golub & Van Loan (2013); Bishop & Nasrabadi (2006). While SVD and PCA aim at reconstructing the data, FA aims at explaining correlations between (questions) items through latent factors Bandalos & Boehm-Kaufman (2010). Factor rotation Browne (2001); Sass & Schmitt (2010); Schmitt & Sass (2011) is then performed to obtain a sparser solution which is easier to interpret and analyze. For a review of FA, see Thompson (2004); Gaskin & Happell (2014); Gorsuch (2014); Goretzko et al. (2021). Non-negative matrix factorization (NMF) was proposed as a way of identifying sparser, more interpretable latent variables, which can be added to reconstruct the data matrix. It was introduced in Paatero & Tapper (1994) and developed in Lee & Seung (2000). Different varieties of NMF-based models have been proposed for various applications, such as the sparsity-controlled Eggert & Korner (2004); Qian et al. (2011), manifold-regularized Lu et al. (2012), orthogonal Ding et al. (2006); Choi (2008), convex/semi-convex Ding et al. (2008), or archetypal regularized NMF Javadi & Montanari (2020). More recently, Deep-NMF Trigeorgis et al. (2016); Zhao et al. (2017) and Deep-MF Xue et al. (2017); Fan & Cheng (2018); Arora et al. (2019) can model non-linearities on top of (non-negative) factors, when the sample is large Fan (2021). These methods do not directly model either the interpretability characteristics or the constraints that we view as desirable.

Obtaining a factorization with these methods requires first specifying the number of latent variables, and then solving an optimization problem. In SVD/PCA, the number of variables is often selected based on the percentage of variance explained, or determined via techniques such as spectral analysis, the Laplace-PCA method, or Velicer's MAP test Velicer (1976); Velicer et al. (2000); Minka (2000). For FA, several methods have been proposed: Bartlett's test Bartlett (1950), parallel analysis Horn (1965); Hayton et al. (2004), MAP test and comparison data Ruscio & Roche (2012). For NMF, iterative detection algorithms are recommended, e.g. the Bayesian information criterion (BIC) Stoica & Selen (2004), cophenetic correlation coefficient (CCC) Fogel et al. (2007) and the dispersion Brunet et al. (2004). More recent proposals for NMF are Bi-cross-validation (BiCV) Owen & Perry (2009) and its generalization, the blockwise-cross-validation (BCV) Kanagal & Sindhwani (2010), which we use in this paper. The optimization problem for NMF is non-convex, and different algorithms for solving it have been proposed. Multiplicative update (MU) Lee & Seung (2000) is the simplest and mostly used. Projected gradient algorithms such as the block coordinate descent Cichocki & Phan (2009); Xu & Yin (2013); Kim et al. (2014) and the alternating optimization Kim & Park (2008); Mairal et al. (2010) aim at scalability and efficiency in larger matrices. Given that our optimization problem has various constraints, we use a combination of alternative optimization and Alternating Direction Method of Multipliers (ADMM) Boyd et al. (2011); Huang et al. (2016).

## 3 Methods

### 3.1 Interpretable Constrained Questionnaire Factorization (ICQF)

**Inputs** Our method operates on a questionnaire data matrix $M \in \mathbb{R}_{\geq 0}^{n \times m}$ with $n$ participants and $m$ questions, where entry $(i, j)$ is the answer given by participant $i$ to question $j$. As questionnaires often have missing data, we also have a mask matrix $\mathcal{M} \in \{0, 1\}^{n \times m}$ of the same dimensionality as $M$, indicating whether each entry is available ($= 1$) or not ($= 0$). Optionally, we may have a confounder matrix $C \in \mathbb{R}_{\geq 0}^{n \times c}$, encoding $c$ known variables for each participant that could account for correlations across questions (e.g. age or sex). We assume these covariates are observed, as standard demographics generally are in questionnaire datasets. If the $j^{th}$ confound $C_{[:,j]}$ is categorical, we convert it to indicator columns for each value. If it is continuous, we first rescale it into $[0, 1]$ (where 0 and 1 are the minimum and maximum in the dataset, or known range), and replace it with two new columns, $C_{[:,j]}$ and $1 - C_{[:,j]}$. This mirroring procedure ensures that monotonic effects in either direction of the confounding variables are representable (e.g. answer patterns more common the younger or the older the participants are), while preserving non-negativity. Importantly $C$ is not used to regress out confounds in the classical linear regression sense. Instead, it acts as a non-negative baseline (offset) components that captures systematic response patterns attributable to known variables. Under the non-negativity constraints, this confound contribution models additive baseline

structure, allowing the latent factors to explain residual variation rather than compensate for it. Lastly, we incorporate a vector of ones into $C$ to model dataset-wide intercept effects, to capture capture global answer tendencies across the population.

**Optimization problem** We seek to factorize the questionnaire matrix $M$ as the product of a $n \times k$ factor matrix $W \in [0,1]$, with the confound matrix $C \in [0,1]$ as optional additional columns, and a $m \times (k+c)$ loading matrix $Q := [^RQ, ^CQ]$, with a loading pattern $^RQ$ over $m$ questions for each of the $k$ factors (and $^CQ$ for optional confounds). Denoting the Hadamard product as $\odot$, our optimization problem minimizes the squared error of this factorization

$$\underset{W \in \mathcal{W}, Q \in \mathcal{Q}, Z \in \mathcal{Z}}{\text{minimize}} \quad 1/2 \left\| \mathcal{M} \odot (M - Z) \right\|_F^2 + \beta \cdot R(W, Q)$$

$$\text{such that} \quad [W, C]Q^T = Z, \ \mathcal{Z} = \{Z | \ \min(M) \leq Z_{ij} \leq \max(M)\}$$

$$\mathcal{Q} = \{Q | \ 0 \leq Q_{ij} \leq b_j\} \ \text{and} \ \mathcal{W} = \{W | \ 0 \leq W_{ij} \leq 1\} \quad \text{(ICQF)}$$

where $b_j$ denotes the upper bound of the response range for question item $j$. [1] The box constraints on $W$, $Q$ and the reconstruction $Z$ are imposed to promote interpretability – loadings become legible as answer patterns rather than arbitrary signed weights – and they additionally stabilize the ADMM updates by confining the iterates to a compact feasible region, which limits extreme rescaling of $W$ and $Q$ and reduces sensitivity to the penalty parameter $\rho$. We also note that these constraints alone do not guarantee uniqueness or full identifiability; their role is to favor interpretable, scale-consistent solutions.

We regularize $W$ and $Q$ through $R(W, Q) := \|W\|_{p,q} + \gamma \|Q\|_{p,q}$ with $\gamma = \frac{n}{m} \max(M)$, where $\|A\|_{p,q} := (\sum_{i=1}^{m} (\sum_{j=1}^{n} |A_{ij}|^p)^{q/p})^{1/q}$. Here, we use $p = q = 1$ to promote sparsity. The scaling factor $\gamma$ normalizes the regularization across factors with different dimensions and value ranges ($W \in [0,1]$ versus $Q \in [\min(M), \max(M)]$), balancing the sparsity penalty between $W$ and $Q$, which otherwise would penalize one matrix far more than the other and potentially hurt interpretability. Note that $\gamma$ does not remove the intrinsic scale ambiguity of matrix factorization; it selects what is, in our view, a more balanced representative from the family of scale-equivalent solutions. Accordingly, we use a single regularization parameter $\beta$ to control the overall sparsity level, treating $\gamma$ as a fixed normalization constant absorbed into the penalty of $Q$.

The sparsity penalty on $Q$ serves two purposes specific to questionnaire factorization. First, it favors a conservative solution in which each factor loads on as few items as is useful for reconstruction, tending to isolate co-occurring core symptom groups. Second, as a consequence, symptom groups may be split across more factors than in factor analysis; such splits are often easier to interpret clinically, since loading magnitudes are comparable across items and can be read as a data-driven refinement of existing questionnaire subscales. Identifying such splits, where they exist in a clinical population, was one of our motivations for the method.

Although ICQF is presented as a constrained factorization rather than a probabilistic model, it encodes a clear generative view. Each latent factor is present in a participant to a degree between 0 (absent) and 1 (fully present); the number of factors simultaneously present is learned from data rather than fixed, with an implicit prior favoring few. Because nonnegativity rules out mutual cancellation, factors combine only additively across items, yielding a parts-based representation. The loadings on the confound columns act as baseline item rates for the subgroup sharing that confound. Like exploratory factor analysis, the formulation remains a linear factorization, but its constraints are chosen to improve identifiability and interpretability rather than to instantiate a richer generative model.

## 3.2 Solving the optimization problem

We use the ADMM framework for fitting the ICQF model, due to its parallelizability, flexibility in incorporating various types of constraints, and its compatibility with different optimization schemes. Specifically, we utilize the Fast Iterative Shrinkage Thresholding Algorithm (FISTA) to accommodate our sparsity

---

[1] We write the bound as item-specific ($Q_{*j} \leq b_j$ in MATLAB-style notation) because in practice questionnaire response ranges are item-specific rather than subject-specific, although the formulation also admits entry-specific bounds $b_{ij}$ in the general case.

constraints, leveraging its numerical advantages, such as quadratic convergence and low memory cost, as discussed in Gaines et al. (2018). Unlike stochastic optimization approaches, which require addressing the missing entries and uneven distribution of responses in questionnaires when generating training batches, ADMM allows us to tackle the optimization problem holistically. Additionally, it can find a solution for large clinical questionnaire datasets – thousands of participants, tens to hundreds of questions – in about a minute with a laptop CPU, our ballpark for practical usefulness. For smaller-scale problems or validation experiments, we also provide a coordinate descent–based optimization procedure (Appendix B), which relies on inexpensive per-coordinate updates and has a smaller memory footprint, and is primarily intended for modest problem sizes.

**Optimization procedure** The ICQF problem is non-convex and requires satisfying multiple constraints. Under the ADMM optimization procedure, the Lagrangian $\mathcal{L}_\rho$ is:

$$\mathcal{L}_\rho(W, Q, Z, \alpha_Z) = 1/2 \|\mathcal{M} \odot (M - Z)\|_F^2 + \mathcal{I}_\mathcal{W}(W) + \beta \|W\|_{1,1} + \mathcal{I}_\mathcal{Q}(Q) + \beta \|Q\|_{1,1}$$
$$+ \langle \alpha_Z, Z - [W, C]Q^T \rangle + \rho/2 \left\| Z - [W, C]Q^T \right\|_F^2 + \mathcal{I}_\mathcal{Z}(Z)$$

where $\rho$ is the penalty parameter, $\alpha_Z$ is the vector of Lagrangian multipliers and $\mathcal{I}_\mathcal{X}(X) = 0$ if $X \in \mathcal{X}$ and $\infty$ otherwise. We update primal variables $W, Q$ and the auxiliary variable $Z$, in alternation, by solving the following sub-problems:

$$W^{(i+1)} = \underset{W \in \mathcal{W}}{\arg\min} \ \rho/2 \|Z^{(i)} - [W, C]Q^{(i),T} + \rho^{-1}\alpha_Z^{(i)}\|_F^2 + \beta \|W\|_{1,1} \tag{1}$$

$$Q^{(i+1)} = \underset{Q \in \mathcal{Q}}{\arg\min} \ \rho/2 \|Z^{(i)} - [W^{(i+1)}, C]Q^T + \rho^{-1}\alpha_Z^{(i)}\|_F^2 + \beta \|Q\|_{1,1} \tag{2}$$

$$Z^{(i+1)} = \underset{Z \in \mathcal{Z}}{\arg\min} \ \|\mathcal{M} \odot (M - Z)\|_F^2 + \rho \|Z - [W^{(i+1)}, C]Q^{(i+1),T} + \rho^{-1}\alpha_Z^{(i)}\|_F^2 \tag{3}$$

for some penalty parameter $\rho$. Lastly, $\alpha_Z$ is updated via

$$\alpha_Z^{(i+1)} \leftarrow \alpha_Z^{(i)} + \rho(Z^{(i+1)} - [W^{(i+1)}, C](Q^{(i+1)})^T) \tag{4}$$

Equations 1 and 2 can be further split into row-wise constrained Lasso problems, which can be solved efficiently via an inner-loop ADMM scheme. This is helpful for dealing with large-scale datasets, if memory is limited, and for acceleration if distributed computational resources are available. For small to moderate problem sizes, coordinate descent (CD) is empirically more efficient and simpler to implement. We briefly outline the CD updates for 1; updates for 2 follows analogously. After rescaling and simplifying notation, the subproblem reduces to

$$\min_{W \in \mathcal{W}} \ \frac{1}{2}\|V - WH\|_F^2 + \frac{\beta_W}{\rho}\|W\|_{1,1}, \tag{5}$$

where $V$ and $H$ are fixed. CD updates one entry $W_{ir}$ at a time by solving a one-dimensional quadratic subproblem, yielding the closed-form update

$$W_{ir} \leftarrow \Pi_{[0, B_W]}\left(W_{ir} - \frac{(WHH^\top - VH^\top)_{ir} + \beta_W/\rho}{(HH^\top)_{rr}}\right), \tag{6}$$

where $\Pi_{[0, B_W]}(\cdot)$ denotes projection onto the feasible interval. Note that missing entries in $M$ do not affect the updates, since the auxiliary variable $Z$ serves as a fully observed surrogate within ADMM. For full derivations and implementation, see Appendix A.

There is a closed form solution for equation 3, since both terms in equation 3 are in Frobenius-norm, so $Z$ can be optimized entry-wise. Specifically, we have the following closed-form solution for $Z^{(i+1)}$:

$$Z^{(i+1)} = \underset{[\min(M), \max(M)]}{Proj}\left(\mathcal{M} \odot M + \rho[W^{(i+1)}, C](Q^{(i+1)})^T - \alpha_Z^{(i)}\right) \oslash (\rho\mathbb{1} + \mathcal{M}) \tag{7}$$

where $\mathbb{1}$ is a 1-matrix with appropriate dimension and $\oslash$ is the Hadamard division. The optimization details within each step are further discussed in Appendix A. Given the flexibility of ADMM, a similar procedure can also be used with other regularizations, if desired.

**Convergence of the optimization procedure**    The convergence hinges on the careful selection of the penalty parameter $\rho$. Informally, imposing the constraint $\rho \geq \sqrt{2}$ on the penalty parameter $\rho$ guarantees monotonicity of the optimization procedure, and that it will converge to a *local* minimum. Integrating this constraint with the adaptive selection of $\rho$ Xu et al. (2017), we obtain an efficient optimization procedure for ICQF. Formally, this can be stated as the following proposition.

**Proposition 3.1** (Non-increasing property). Assuming $\rho \geq \sqrt{2}$, we have

$$0 \leq \mathcal{L}_\rho(W^{(i+1)}, Q^{(i+1)}, Z^{(i+1)}, \alpha_Z^{(i+1)}) \leq \mathcal{L}_\rho(W^{(i)}, Q^{(i)}, Z^{(i)}, \alpha_Z^{(i)}) \quad \forall i. \tag{8}$$

and by the monotone convergence theorem, $(W^{(i)}, Q^{(i)})$ will converge to a critical point $(W, Q)$.

**Proof Sketch:** We present a sketch of the proof establishing the monotonicity property of the augmented Lagrangian in Equation 8. Let $\mathbb{V}^{(i)} = \{W^{(i)}, Q^{(i)}, Z^{(i)}\}$ and define $R^{(i)} = [W^{(i)}, C](Q^{(i)})^T$. We examine the difference in the augmented Lagrangian values between two consecutive iterations,

$$\mathcal{L}_\rho(\mathbb{V}^{(i+1)}, \alpha_Z^{(i+1)}) - \mathcal{L}_\rho(\mathbb{V}^{(i)}, \alpha_Z^{(i)}) = \underbrace{\mathcal{L}_\rho(\mathbb{V}^{(i+1)}, \alpha_Z^{(i+1)}) - \mathcal{L}_\rho(\mathbb{V}^{(i+1)}, \alpha_Z^{(i)})}_{(I)}$$

$$+ \underbrace{\mathcal{L}_\rho(\mathbb{V}^{(i+1)}, \alpha_Z^{(i)}) - \mathcal{L}_\rho(\mathbb{V}^{(i)}, \alpha_Z^{(i)})}_{(II)}. \tag{9}$$

Using the ADMM dual update $\alpha_Z^{(i+1)} = \alpha_Z^{(i)} + \rho(Z^{(i+1)} - R^{(i+1)})$, term $(I)$ can be written as

$$(I) = \langle \alpha_Z^{(i+1)} - \alpha_Z^{(i)}, Z^{(i+1)} - R^{(i+1)} \rangle = \frac{1}{\rho} \|\alpha_Z^{(i+1)} - \alpha_Z^{(i)}\|_F^2. \tag{10}$$

We further decompose term $(II)$ according to the sequential updates of $Z, Q$, and $W$,

$$(II) = \underbrace{\mathcal{L}_\rho(\mathbb{V}^{(i+1)}, \alpha_Z^{(i)}) - \mathcal{L}_\rho(\mathbb{V}^{(i+1,i+1,i)}, \alpha_Z^{(i)})}_{(\mathcal{A})} + \underbrace{\mathcal{L}_\rho(\mathbb{V}^{(i+1,i+1,i)}, \alpha_Z^{(i)}) - \mathcal{L}_{\rho(\mathbb{V}^{(i+1,i,i)}, \alpha_Z^{(i)})}}_{(\mathcal{B})}$$

$$+ \underbrace{\mathcal{L}_\rho(\mathbb{V}^{(i+1,i,i)}, \alpha_Z^{(i)}) - \mathcal{L}_\rho(\mathbb{V}^{(i)}, \alpha_Z^{(i)})}_{(\mathcal{C})}. \tag{11}$$

The term $(\mathcal{A})$ corresponds to the $Z$-update. By expanding the quadratic terms in the augmented Lagrangian and using the optimality of $Z^{(i+1)}$ for the $Z$-subproblem, which is strongly convex, we obtain

$$(\mathcal{A}) \leq -\frac{\rho}{2} \|Z^{(i+1)} - Z^{(i)}\|_F^2. \tag{12}$$

The term $(\mathcal{B})$ captures the effect of the $Q$-update. Each row of $Q^{(i+1)}$ is obtained by solving a constrained Lasso problem. The first-order optimality condition and the convexity of the $\ell_1$-norm imply

$$(\mathcal{B}) \leq -\frac{\rho}{2} \|[W^{(i+1)}, C](Q^{(i+1),T} - Q^{(i),T})\|_F^2. \tag{13}$$

Similarly, the contribution of the $W$-update satisfies

$$(\mathcal{C}) \leq -\frac{\rho}{2} \|[(W^{(i+1)} - W^{(i)}), C]Q^{(i),T}\|_F^2. \tag{14}$$

To bound the dual increment, we use the definition of $R^{(i)}$ and obtain

$$\|\alpha_Z^{(i+1)} - \alpha_Z^{(i)}\|_F^2 \leq \|Z^{(i+1)} - Z^{(i)}\|_F^2 + \|[W^{(i+1)}, C](Q^{(i+1),T} - Q^{(i),T})\|_F^2$$

$$+ \|[(W^{(i+1)} - W^{(i)}), C]Q^{(i),T}\|_F^2. \tag{15}$$

Combining the above bounds yields

$$\mathcal{L}_\rho(\mathbb{V}^{(i+1)}, \alpha_Z^{(i+1)}) - \mathcal{L}_\rho(\mathbb{V}^{(i)}, \alpha_Z^{(i)}) \leq \left(\frac{1}{\rho} - \frac{\rho}{2}\right) \cdot (\text{nonnegative terms}), \tag{16}$$

which implies that the augmented Lagrangian is non-increasing whenever $\rho \geq \sqrt{2}$. The full proof of Proposition 3.1 is given in Appendix C. $\qquad\square$

Furthermore, Bjorck et al. (2021) showed that, for non-negative matrix factorizations, if the dimensionality $k$ is the same as that $k^*$ of a ground truth solution $(W^*, Q^*)$, the error $\|M - WQ^T\|_F^2$ is star-convex towards $(W^*, Q^*)$, and the solution is close to a *global* minimum. However, if $k \neq k^*$, the relative error between $W^*$ and $W$ increases with $|\sqrt{k/k^*} - 1|$. Inaccurate estimation of $k^*$ thus affects both the interpretability of $(W, Q)$ and the convergence to global minima. With the bounded constraints imposed on $W$ and $Q$ in ICQF, Popoviciu's inequality establishes an upper bound for the variances $\sigma_W^2$ and $\sigma_Q^2$ of each column in $W$ and $Q$ respectively. To simplify the analysis, we assume equal variances among the columns (generally true). Then we have the following proposition:

**Proposition 3.2.** Let $(W^*, Q^*)$ be a ground-truth factorization of the given $\mathbf{M} = \mathbf{W}^*(\mathbf{Q}^*)^T$, with latent dimension $k^*$, where $\mathbf{W}^*$ and $\mathbf{Q}^*$ are matrix-valued random variables with entries sampled from bounded distributions. Suppose $(\mathbf{W}, \mathbf{Q})$ be another factorization with latent dimension $k \neq k^*$ that achieves the same expected reconstruction error and the same expected approximation to $\mathbf{M}$, and whose latent coordinates are isotropic in expectation. Then, with high probability,

$$\mathbb{E}\left[\|\mathbf{W}^* - \mathbf{W}\|_F^2\right] \geq \left(\sqrt{k/k^*} - 1\right)^2 \mathbb{E}\left[\|\mathbf{W}^*\|_F^2\right] \tag{17}$$

The full proof of Proposition 3.2 is provided in Appendix D.

The two propositions, combined, show that our factorization can capture the true latent structure of the data, under the right conditions. The first assumption is that $M = W^*(Q^*)^\top$ is a good approximation, i.e. the responses are well represented by a linear combination of latent factors, which is reasonable for questionnaire data since responses are typically viewed as arising from a small number of latent constructs the questionnaire was designed to measure. The second is having a robust estimator of the latent dimension $k^*$. We use blockwise cross-validation (BCV) for this purpose.

**Choice of number of factors** Given a matrix $M$, for each possible $k$ considered, we permute the rows and columns of $M$. When confound variables $C$ are available, the row permutation can also be performed in a stratified manner, so as to preserve the distribution of $C$. The permuted matrix is then partitioned into $b_r \times b_c$ blocks, each of which is assigned into one of 10 folds. We then carry out a loop over folds where, for each fold, we 1) omit the blocks in that fold from the matrix, 2) factorize the remainder of the matrix, 3) impute the omitted blocks via matrix completion, and 4) compute the reconstruction error [2] over omitted blocks. We choose $k$ with the lowest average error over folds. We compared this with other approaches for choosing $k$, for ICQF and other methods, over synthetic data, and report the results in Section 4.1.

## 4 Experiments and results

Across experiments, we compare ICQF with the methods most likely to be used in factorizing questionnaires. Our first baseline method is $\ell_1$- regularized NMF ($\ell_1$-NMF) Cichocki & Phan (2009), as it also imposes non-negativity and sparsity constraints. As constructs (or questions) can be correlated, we rule out other NMF methods with orthogonality constraints. FA with promax rotation (FA-promax) Hendrickson & White (1964) using minimum residual as estimation method is included because it is the most commonly used technique for analyzing questionnaires and extracting latent constructs. It is also a baseline familiar to the clinical community designing questionnaires. Both ICQF and $\ell_1$-NMF were initialized with NNDSVD Boutsidis & Gallopoulos (2008), and the same stopping criterion (relative iteration convergence tolerance $\epsilon < 1e{-}3$) for fairness. In experiments without hyperparameter tuning, both methods use the same fixed sparsity strength

---

[2]A multiplicative correction factor is applied to the error if number of blocks in the last fold is less than in others.

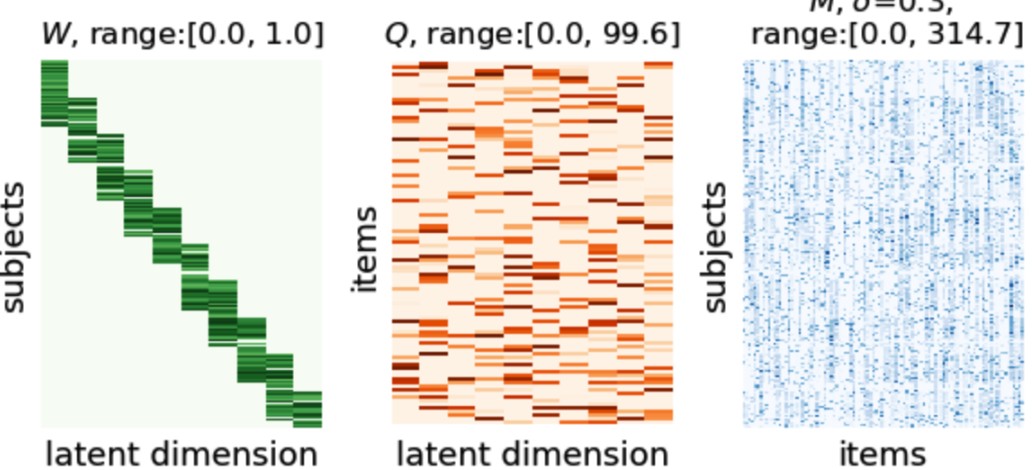

Figure 1: Synthetic $W$, $Q$ and $M$ with $\delta = 0.3$.

($\beta = 1e{-}1$) as a common setting. When tuning is performed, BCV selects not only $k$ but also $\beta$ for each method, by minimizing the average left-out error jointly over a grid of $(k, \beta)$ pairs (e.g. Appendix F), so that no method is advantaged by a hand-picked penalty.

## 4.1 Experiments on synthetic questionnaire data

Our first experiment was aimed at examining the effectiveness of BCV and other algorithms at estimating the number of latent factors when paired with the three factorization methods. To that effect we generated a realistic synthetic questionnaire with $k^* = 10$ factors as ground truth. To generate a $200 \times 10$ latent factor matrix $W$ (Figure 1 left), we first defined a matrix $D$, a binary matrix in which columns are correlated in a step-like pattern. Each column is present in isolation for 20 participants, and in tandem with another for 10 more, to synthesize correlation between factors. Then an entry of $W[i, j]$ is defined as

$$W[i, j] := D[i, j] \cdot a \cdot b, \quad a \sim U(0.5, 1), \ b \sim B(1, 0.9) \tag{18}$$

where $U(0.5, 1)$ is Uniform in $[0.5, 1]$ and $B(1, 0.9)$ is Bernoulli with probability $p = 0.9$. Since this experiment targets recovery of the latent dimension $k^*$ rather than inference of $W$ on new data, we use the full $W$ (200 samples) as the training set and, for each method, estimate $k$ and compare it against $k^*$. We note that a held-out factor matrix $W^{\text{held-out}}$ could equally be drawn from the same designed distribution as $W$ (Equation 18) under an 80/20 split, which is the setting used in our later prediction experiments.

Each factor had an associated loading vector – answer pattern – over 100 questions ($[0, 100]$ range). The resulting $100 \times 10$ loading matrix $Q$, shown in Figure 1 (center), was defined to be

$$Q[i, j] := c \cdot d, \quad c \sim U(0, 100), \ d \sim B(1, 0.3) \tag{19}$$

We then created a noiseless data matrix $M_{clean} := \min(0, \max(WQ^T, 100))$, and added noise by

$$M := \min\left(0, \max(M_{clean} + e \cdot f, 100)\right), \quad f \sim U(-100, 100) \tag{20}$$

where $e$ follows a discrete probability distribution with $P(e = 1) = \delta, P(e = 0) = 1 - \delta$. This yielded a data matrix $M$, shown in Figure 1 (right) for $\delta = 0.3$ (the highest noise level) [3]. Table 1 shows the mean error $\bar{\epsilon}$ and the standard error $s_E$ of the detected $k$ versus ground-truth $k^* = 10$, across 30 generated datasets. We tested five popular detection algorithms: BCV Kanagal & Sindhwani (2010), $BIC_1$ Stoica & Selen (2004)[4],

---

[3]Full version figure with an factorization result is reported in Appendix G.

[4]Here $BIC_1(k) := \log\left(\|M - WQ^T\|_F^2\right) + k\frac{m+n}{mn}\log\left(\frac{mn}{m+n}\right)$, other versions yield similar results.

| Detection schemes for $k$ | Noise density $\delta$ | | |
|---|---|---|---|
| | $\delta = 0.1$ | $\delta = 0.2$ | $\delta = 0.3$ |
| ICQF (BCV) | $(0.10, 0.06)$ | $(0.11, 0.06)$ | $(0.77, 0.15)$ |
| $\ell_1$-NMF (BCV) | $(0.17, 0.07)$ | $(2.37, 0.33)$ | $(2.40, 0.31)$ |
| ICQF ($BIC_1$) | $(0.10, 0.06)$ | $(0.67, 0.23)$ | $(2.40, 0.48)$ |
| $\ell_1$-NMF ($BIC_1$) | $(0.90, 0.23)$ | $(1.10, 0.30)$ | $(2.47, 0.54)$ |
| ICQF (CCC) | $(1.33, 0.24)$ | $(1.14, 0.21)$ | $(0.96, 0.18)$ |
| $\ell_1$-NMF (CCC) | NaN | NaN | NaN |
| ICQF (Dispersion) | $(0.23, 0.09)$ | $(0.93, 0.16)$ | $(2.60, 0.26)$ |
| $\ell_1$-NMF (Dispersion) | NaN | NaN | NaN |
| FA-promax (PA) | $(0.17, 0.07)$ | $(0.53, 0.10)$ | $(0.87, 0.14)$ |
| FA-promax (MAP) | $(0.11, 0.06)$ | $(0.13, 0.06)$ | $(1.27, 0.20)$ |
| FA-promax ($BIC_2$) | $(0.30, 0.03)$ | $(0.93, 0.11)$ | NaN |

Table 1: Average error and standard error $(\bar{\epsilon}, s_E)$ of $k$.

CCC Fogel et al. (2007) and Dispersion Brunet et al. (2004). For ICQF and $\ell_1$-NMF, BCV was the best detection scheme at all noise levels; $BIC_2$ performed well for low noise only. For the three common FA schemes, Horn's PA Horn (1965) and MAP Velicer (1976) were superior to $BIC_2$ Preacher et al. (2013), which aligns with empirical observations in Velicer et al. (2000); Watkins (2018); Goretzko et al. (2021). ICQF with BCV outperformed $\ell_1$-NMF and FA at all noise levels.

## 4.2 Experiments with the Child Behavior Checklist (*CBCL*) questionnaire

### 4.2.1 Data

The 2001 Child Behavior Checklist (*CBCL*) is a questionnaire covering different domains of psychopathology, designed to screen patients for referral to pediatric psychiatry clinics, for a variety of diagnoses Heflinger et al. (2000); Biederman et al. (2005; 2020).The checklist includes 113 questions, grouped into 8 syndrome subscales: *Aggressive, Anxiety/Depressed, Attention, Rule Break, Social, Somatic, Thought, Withdrawn* problems. The referral is typically based on scores obtained by adding answers on syndrome-specific subscales. Answers are scored on a three-point Likert scale (0=absent, 1=occurs sometimes, 2=occurs often) and the time frame for the responses is the past 6 months. We use the parent-reported CBCL responses.

The primary experiments in this paper use CBCL questionnaires from two independent studies: the Healthy Brain Network (*HBN*) Alexander et al. (2017) and the Adolescent Brain Cognitive Development[SM] (*ABCD*) study (https://abcdstudy.org). HBN is an ongoing project to create a biobank from NYC area care-seeking children and adolescents. ABCD is a longitudinal study, starting with youths aged 9-10, to obtain a socio-demographically representative sample over time. Both datasets provide diagnostic labels for mental health conditions, of which we selected the 11 most prevalent ones (Depression, General Anxiety, ADHD, Suspected ASD, Panic, Agoraphobia, Separation and Social Anxiety, BPD, Phobia, OCD, Eating Disorder, PTSD, Sleep problems). In HBN, we use CBCL from 1335 participants, 1,001 of whom have at least one diagnosis. In ABCD, we use CBCL from 11,681 participants, 7,359 of whom have at least one diagnosis.

### 4.2.2 Experimental setup

**Baseline methods** As explained above, we use $\ell_1$-regularized NMF ($\ell_1$-NMF) Cichocki & Phan (2009) and factor analysis with promax rotation Hendrickson & White (1964) as baseline methods. We also include syndrome subscale scores, since they are often used for diagnostic prediction in screening. To estimate the number of factors $k$, we use BCV for $\ell_1$-NMF and ICQF, and Horn's parallel analysis for FA, the best approach for each method in the synthetic questionnaire experiments in Section 4.1.

**Dataset splits** Within each dataset, we first split the participants into development and held-out test sets with an 80/20 ratio. The assignment is done using stratified sampling, to keep the distribution of confounds and diagnostic labels similar across both sets. Training and validation sets are derived from the development

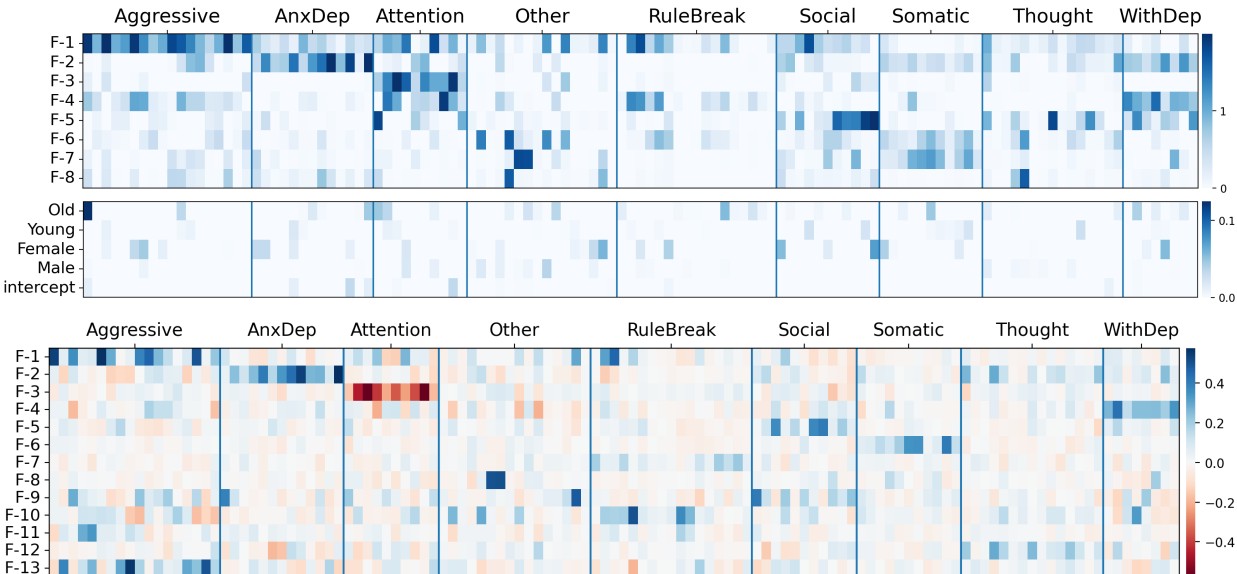

Figure 2: Heatmap of factor loadings $Q := [^R Q, ^C Q]$ from ICQF for factors proper, old/young and male/female confounds, and the implicit intercept (top) and loadings $Q$ from Factor Analysis with promax rotation (bottom). Questions are grouped by syndrome subscale; some factors are syndrome specific, while others bridge syndromes. For the enlarge version, see Appendix H.

set, as explained in each experiment, and used to set all the parameters for testing. All the quantitative results are obtained on the held-out set. To increase the robustness of our analysis, and obtain measures of uncertainty, we use different seeds to resample 30 dataset splits, and carry out experiments on each split. The reported results are obtained by averaging the results on the held-out set across all 30 splits.

**Model training and inference** Let $W^{\text{set}}$ denote the participant factor matrix in ICQF or NMF, or the analogous factor score matrix in FA, with the superscript denoting the dataset. Similarly, let $Q$ denote the question loadings associated with a factor in each method. Model training will yield a $(W^{\text{train}}, Q)$ for participants in the training set. Inference with the model will produce $W^{\text{validate}}$ and $W^{\text{held-out}}$ in validation and held-out sets, using the trained $Q$ and confounds $C^{\text{validate}}, C^{\text{held-out}}$ (if applicable).

### 4.2.3 Experiment 1: qualitative comparison of ICQF with FA

We begin with a qualitative assessment of ICQF applied to the development set portion of the CBCL questionnaire from the HBN dataset. We estimated the latent dimensionality $k = 8$ using BCV to compute an error over left-out data, at each possible $k$. The regularization parameter $\beta = 0.5$ was set the same way. The top-panel of Figure 2 shows the heat map of the loading matrix $Q := [^R Q, ^C Q]$, composed of loadings $^R Q$ for the latent factors $W$, and the loadings $^C Q$ for the confounds $C$.

Given the absence of ground-truth factorizations for this questionnaire, the qualitative assessment hinges on the relation of question loadings to the syndrome subscales used in clinical practice. While there were factors that loaded primarily in questions from one subscale, as expected, we were encouraged by finding others that grouped questions from multiple subscales, in ways that were deemed sensible by our clinical collaborators. As a further check, we inspected the loadings of confound **Old** (increasing age) and observe that they covered issues such as *"Argues", "Act Young", "Swears" and "Alcohol"*. The loadings of $Q$ also reveal the relative importance among questions in each estimated factor; for comparison, subscales would deem all questions equally important. Visualization of each latent factor and their corresponding themes and questions are reported in Appendix I. The themes were determined by our clinical collaborators, taking into consideration the weights each factor assigned across questions.

For comparison, Figure 2 (bottom) shows the loadings $Q$ from Factor Analysis with promax rotation. Using parallel analysis, we identified a value of $k = 13$, which significantly exceeds the 8 syndrome subscales that were initially established during the development of the checklist. The absence of sparsity and non-negativity control also results in a matrix that is more densely populated with both positive and negative elements, in an arbitrary range. This can present challenges when attempting to interpret the loadings in conjunction with the factor matrix $W$, also without constraints. A detailed subjective evaluation by our clinical collaborators, showcasing the improvement of the proposed ICQF against FA, is given in Appendix I.

For a fairer comparison we also computed loadings with $\ell_1$-NMF, for which BCV selected $\beta = 0.0$ and $k = 11$ – no additional $\ell_1$ sparsity was supported by the data. Without a bound on the loading range, a few factors can attain large magnitudes that dominate the heat map. One could tune $k$ down or rescale the columns of $Q$ for a denser display, but such adjustments would no longer be purely data-driven. Since matrix factorization is scale-indeterminate, rescaling $Q$ induces an inverse rescaling of $W$ and alters downstream latent scores. This post hoc transformation, shared with factor analysis, is avoided by ICQF. More broadly, the fact that BCV selected $\beta = 0$ suggests its reconstruction objective need not align with interpretability for $\ell_1$-NMF. Appendix H shows all three methods together.

### 4.2.4  Experiment 2: preservation of diagnostic-related information

Our first quantitative metric to compare ICQF with baseline methods is the degree to which the low-dimensional factor representation of each participant (row of $W$) retains diagnostic information, across all 11 conditions we consider. Furthermore, this metric must be evaluated as a function of training sample size. As typical clinical studies have sample sizes an order of magnitude smaller than these, the regularization imposed by each method becomes more influential in determining the relationship between questions.

We evaluate this by creating training sets of different sizes from the development set (80, 60, 40, and 20 % of participants, with a fixed 20% as a validation set) and factorizing each of them with ICQF and the other methods. This yields a $W^{\text{train}}$, $Q^{\text{train}}$ for each combination of method and training set size. The decomposition is fit on training data alone; to obtain scores for held-out subjects we fix the trained $Q$ and optimize the ICQF objective over $W$ only (Appendix E), yielding $W^{\text{held-out}}$. The test data are never used to learn $Q$, which avoids information leakage into the prediction pipeline. The same held-out set is used for *every* method and dataset size being compared.

To estimate diagnostic prediction performance for each $W^{\text{train}}$, $Q^{\text{train}}$ factorization, we train a separate logistic regression model with $\ell_2$ regularization and balanced class weights from $W^{\text{train}}$ for each of the 11 diagnostic labels (i.e., 11 binary classification problems). The regularization strength is fine-tuned using $W^{\text{validate}}$, and prediction assessment is carried out on $W^{\text{held-out}}$ using the receiver operating characteristic (ROC) area under the curve (AUC) metric. The use of AUC is motivated from a clinical perspective, where clinicians often apply varying thresholds for detection depending on downstream use, such as screening versus intervention. We ran this procedure in both CBCL-HBN and CBCL-ABCD data.

Figure 3 shows the trend and variability (95% confidence region) of the average AUCs across 11 diagnostic problems, for ICQF and the baseline methods, using different dataset sizes (proportions of subjects), for HBN (left) and ABCD (right). In both HBN and ABCD, the ICQF outperforms other optimal baseline methods in maintaining high AUC scores across 11 conditions, and the difference in performance increases as the sample size decreases ($p \leq 0.01$, based on a one-side Wilcoxon signed rank test and adjusted using False Discovery Rate $\alpha = 0.01$), except for $\ell_1$-NMF at 20% in CBCL-HBN). Moreover, the factorization solutions obtained with ICQF are more stable in terms of the number of dimensions $k$ ($k = 8 \rightarrow 6$ for ICQF, versus $8 \rightarrow 3$ for $\ell_1$-NMF and $13 \rightarrow 18$ for FA-promax in HBN; $k = 7 \rightarrow 7$ for ICQF, versus $5 \rightarrow 4$ for $\ell_1$-NMF and $20 \rightarrow 17$ for FA-promax in ABCD). This is particularly noteworthy in comparison to $\ell_1$-NMF, as it indicates the extra bounded constraints on $W$ and the approximation matrix $M_{approx}$ makes BCV detect $k$ more consistently.

### 4.2.5  Experiment 3: quality of the factor loadings

Our second quantitative metric to compare ICQF with baseline methods considers the change in quality of the factor loading matrix $Q$ as training sample size decreases, to examine the effect of regularization in

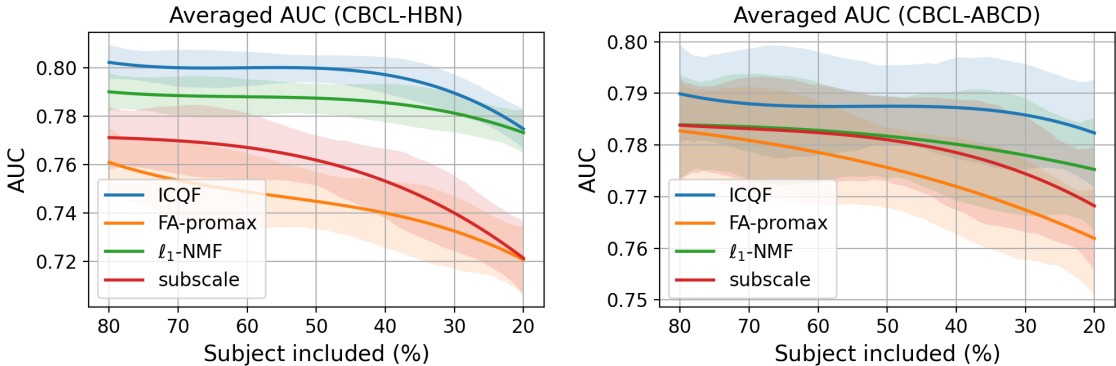

Figure 3: Trend and variability in average diagnostic prediction performance across 11 conditions, using decreasing dataset sizes, in CBCL questionnaires from HBN (left) and ABCD (right) independent datasets.

Table 2: **CBCL-HBN and CBCL-ABCD:** Quality of $Q$ factor loadings at various training set sizes, within dataset. The values are the mean and standard deviation of Pearson correlation coefficients between best-matched $Q$ factors from the full dataset, and from decreasing size subsets of it. Bolded where ICQF is significantly better. **CBCL-HBN ↔ CBCL-ABCD:** Agreement in $Q$ factor loadings between models estimated in CBCL in two independent datasets, measured in the same way.

| | | Factorization | | |
|---|---|---|---|---|
| **Questionnaire** | $n$-subjects | ICQF | FA-promax | $\ell_1$-NMF |
| | 1854 (80%) | **0.89** (0.07) | 0.51 (0.41) | 0.76 (0.18) |
| CBCL-HBN | 1388 (60%) | **0.94** (0.03) | 0.62 (0.34) | 0.75 (0.19) |
| | 924 (40%) | **0.92** (0.05) | 0.62 (0.33) | 0.75 (0.19) |
| | 462 (20%) | 0.85 (0.12) | 0.54 (0.36) | 0.76 (0.20) |
| | 7474 (80%) | **0.84** (0.13) | 0.43 (0.27) | 0.63 (0.28) |
| CBCL-ABCD | 5604 (60%) | **0.84** (0.13) | 0.32 (0.30) | 0.63 (0.28) |
| | 3736 (40%) | **0.77** (0.20) | 0.42 (0.24) | 0.63 (0.28) |
| | 1868 (20%) | **0.69** (0.25) | 0.35 (0.26) | 0.62 (0.29) |
| CBCL-HBN ↔ CBCL-ABCD | full ↔ full | **0.75 (0.07)** | 0.71 (0.03) | 0.68 (0.08) |

constraining estimates. Ideally, a method should produce similar loadings across questions for each factor even as training datasets get smaller. As before, we obtain a $W^{\text{train}}$, $Q^{\text{train}}$ for each combination of method and training set size. We then compare the loading matrix at each size ($Q_\%$) with the one obtained on the full development dataset ($Q_{\text{full}}$). We do this by greedily matching each row from $Q_{\text{full}}$ with a row from $Q_\%$ by their Pearson correlation, and then computing the average correlation across pairs as the score. Given that a factorization learned on a smaller dataset may have fewer factors, we do this over the first $\min(k_{\text{full}}, k_\%)$ rows only. The first two rows of Table 2 reports this score for ICQF and the two baseline factorization methods, at each dataset size, on both CBCL-HBN and CBCL-ABCD datasets. ICQF outperforms the other methods at every dataset size ($p \leq 0.01$, based on a one-side Wilcoxon signed rank test and adjusted using False Discovery Rate $\alpha = 0.01$), except for $\ell_1$-NMF at 20% in CBCL-HBN.

Our third quantitative metric is the replicability of factor loadings across independent studies (and populations). This is an important criterion for clinical research purposes, as it means that the relations between questions identified by the factorization are general. We measure this by computing $W, Q$ for the full development sets of HBN and ABCD, for ICQF and the two baseline factorization methods. For each method, we greedily match factors loadings for the HBN and ABCD factorizations, and compute the average Pearson correlation across factor pairs, reported on the third row of Table 2. We conduct similar statistical testing and observe that ICQF outperforms the other methods ($p \leq 0.05$).

# 5 Discussion

In this paper, we introduced ICQF, a non-negative matrix factorization method designed for questionnaire data. Our method incorporates characteristics that enhance the interpretability of the resulting factorization, as conveyed by our clinical research collaborators and co-authors. We showed that their qualitative desiderata can be turned into formal constraints in the factorization problem, together with direct modelling of confounding variables, which other methods do not allow. The method is user friendly, by supporting automated estimation of the number of factors, minimizing the number of hyper-parameters, and transparently handling missing entries instead of requiring separate imputation. The characteristics above mean that ICQF required an entire optimization procedure to be derived from scratch. We provided a theoretical formalization of the problem and the procedure, and demonstrated a pair of propositions that guarantee convergence of the procedure to a local minimum and, in certain conditions, a global minimum as well.

We evaluated ICQF against alternative methods for the same purpose ($\ell_1$-NMF, used in the machine learning literature, and factor analysis, used in the clinical literature), on a widely used clinical questionnaire, in participants from two completely independent datasets. We designed metrics capturing the desired properties, namely preservation of diagnostic information – as this questionnaire is used for screening – and stability of solutions, at a range of dataset sizes, or across independent datasets. We carried out experiments controlling these factors, and showed that ICQF outperforms the alternative methods across the board. We have also used ICQF with 20 other questionnaires in HBN – both general-purpose and disorder-specific – in experiments not reported in this paper, and the results are generally sensible from the perspective of clinical research collaborators. Overall, results suggest that the regularization imposed by ICQF matches the underlying characteristics of questionnaire data better than other methods, in addition to promoting interpretability.

Finally, ICQF opens up two research directions in psychiatry, which we are currently exploring. The first is the identification of relationships between latent constructs estimated from different questionnaires. This can be implemented through a first stage ICQF factorization of each questionnaire, and a second stage *meta-factorization*, a ICQF factorization of the matrix obtained by concatenating the latent factors estimated in the first stage. Preliminary experiments indicate that this approach yields clinically plausible meta-factors, and also provides a principled way to select a compact set of items, across all questionnaires, that suffice to estimate them with minimal redundancy. This could be the basis for the design of novel broad questionnaires that could efficiently estimate the presence of the full range of latent constructs considered here. The second direction is *longitudinal factorization*, i.e. the identification of latent factors whose item associations may vary across ages. This would allow modeling of the empirical phenomenon of a given diagnostic being associated with different symptoms through development. To use ICQF for this purpose, it suffices to array the longitudinal data as a matrix with one row per participant and $\#items \times \#age_bins$ columns, which can then be factored. Each latent factor would be associated with loadings over items, at each age (or age bin). As participants likely would not have answered questionnaires at every possible age, the matrix would have many entries missing by design, as well as those missing due to nonresponse or incomplete follow-up. Missing data can be seamlessly handled by ICQF, in general, and potentially even in different ways depending on whether it happens by design or not. For these reasons, we believe ICQF may enable new directions of research in psychiatry, in addition to already being of broad interest to researchers analyzing questionnaire data.

**Broader Impact Statement**

We do not anticipate a negative impact of this work. Any factorizations of questionnaire data obtained with this method would be evaluated quantitatively (if used for downstream prediction tasks) and qualitatively (for various psychiatric validity criteria).

**Acknowledgments**

This research was supported in part by the Intramural Research Program of the National Institutes of Health (NIH), ZIC-MH002968 (KL and FP) and ZIA-MH002949 (BM and AR). The contributions of the NIH author(s) are considered Works of the United States Government. The findings and conclusions presented

in this paper are those of the author(s) and do not necessarily reflect the views of the NIH or the U.S. Department of Health and Human Services. The authors would like to thank Dylan Nielson for valuable feedback on the paper, the method, and its implementation in the publicly available code package.

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

## Appendix

**Content**

## A  Optimization procedure of ICQF

Recall that the Lagrangian $\mathcal{L}_\rho$ of ICQF is:

$$\mathcal{L}_\rho(W, Q, Z, \alpha_Z) = \frac{1}{2}\|\mathcal{M} \odot (M - Z)\|_F^2 + \mathcal{I}_{\mathcal{W}}(W) + \beta\|W\|_{1,1} + \mathcal{I}_{\mathcal{Q}}(Q)$$
$$+ \beta\|Q\|_{1,1} + \langle \alpha_Z, Z - [W, C]Q^T \rangle + \frac{\rho}{2}\left\|Z - [W, C]Q^T\right\|_F^2 + \mathcal{I}_{\mathcal{Z}}(Z)$$

Following the ADMM approach, we alternatingly update primal variables $W, Q$ and the auxiliary variable $Z$, instead of updating them jointly. In particular, we iteratively solve the following sub-problems:

$$W^{(i+1)} = \underset{W \in \mathcal{W}}{\arg\min} \ \frac{\rho}{2}\left\|Z^{(i)} - [W, C]Q^{(i),T} + \frac{1}{\rho}\alpha_Z^{(i)}\right\|_F^2 + \beta\|W\|_{1,1} \qquad \text{(Sub-problem 1)}$$

$$Q^{(i+1)} = \underset{Q \in \mathcal{Q}}{\arg\min} \ \frac{\rho}{2}\left\|Z^{(i)} - [W^{(i+1)}, C]Q^T + \frac{1}{\rho}\alpha_Z^{(i)}\right\|_F^2 + \beta\|Q\|_{1,1} \qquad \text{(Sub-problem 2)}$$

$$Z^{(i+1)} = \underset{Z \in \mathcal{Z}}{\arg\min} \ \frac{1}{2}\|\mathcal{M} \odot (M - Z)\|_F^2 + \frac{\rho}{2}\left\|Z - [W^{(i+1)}, C]Q^{(i+1),T} + \frac{1}{\rho}\alpha_Z^{(i)}\right\|_F^2 \qquad \text{(Sub-problem 3)}$$

for some penalty parameter $\rho$. We denote the Hadamard product as $\odot$. The vector of Lagrangian multipliers $\alpha_Z$ is updated via

$$\alpha_Z^{(i+1)} \leftarrow \alpha_Z^{(i)} + \rho(Z^{(i+1)} - [W^{(i+1)}, C](Q^{(i+1)})^T) \tag{21}$$

**Sub-problems 1 and 2 (Equations 1 and 2)**

Note that equation 1 (and similarly equation 2 by taking the transpose) can be split into row-wise constrained Lasso problem. Specifically, the $r^{\text{th}}$ row problem can be simplified into:

$$x^* = \arg\min_{0 \leq x_i \leq 1} \frac{\rho}{2}\|b - Ax\|_F^2 + \beta\|x\|_1, \quad A = Q^{(i)}, \ b = \left[Z^{(i)} - CQ^{(i),T} + \frac{1}{\rho}\alpha_Z^{(i)}\right]_{[r,:]} \tag{22}$$

Here we use the Matlab matrix notation $[\cdot]_{[r,:]}$ to represent row extraction operation. As suggested in Gaines et al. (2018) one can also use ADMM to solve equation 22:

$$x^{(i+1)} = \arg\min_{x} \frac{\rho}{2}\|b - Ax\|_2^2 + \frac{\tau}{2}\|x - y^{(i)} + \frac{1}{\tau}\mu^{(i)}\|_2^2 + \beta\|x\|_1 \tag{23}$$

$$y^{(i+1)} = Proj_{[0,1]}(x^{(i+1)} + \frac{1}{\tau}\mu^{(i)}) \tag{24}$$

$$\mu^{(i+1)} \leftarrow \mu^{(i)} + \tau(x^{(i+1)} - y^{(i+1)}) \tag{25}$$

Similarly, $\mu$ is the vector of Lagrangian multipliers and $\tau$ is the penalty parameter. $Proj_{[0,1]}$ refers to the orthogonal projection into $[0,1]$ (inherited from the box-constraints of $W$). Equation 23 can be solved via the well-established FISTA algorithm Beck & Teboulle (2009). Consider the following optimization problem

$$\arg\min_{x} \ \lambda\|x\|_1 + \frac{1}{2}f(x) \tag{26}$$

The FISTA algorithm for solving 26 is summarized as follows:

---
**Algorithm 1:** FISTA for equation 26

---
**Initialize:** $\delta = 1\mathrm{e}{-6}$; $x_{-1} = \mathbf{0}, x_0 = t_0 = \mathbf{1}$
**Input:** $L$, Lipschitz constant of $\nabla f$
**Result:** Solution $x$ of equation 26
**while** $\|x_i - x_{i-1}\|_2 > \delta$ **do**
  $\widetilde{x}_{i+1} = \arg\min_z \left\{\frac{\lambda}{L}\|z\|_1 + \frac{1}{2}\left\|z - \left(x_i - \frac{1}{L}\nabla f(x_i)\right)\right\|\right\}$;
  $t_{i+1} = \frac{\mathbf{1}+\sqrt{1+4t_i^2}}{2}$;
  $x_{i+1} = \widetilde{x}_{i+1} + \frac{t_i-\mathbf{1}}{t_{i+1}}(\widetilde{x}_{i+1} - x_i)$;
**end**

---

To solve equation 23 with FISTA algorithm, using the notation as introduced in equation 22, we have

$$f(x) = \rho\|b - Ax\|_2^2 + \tau\|x - y^{(i)} + \frac{1}{\tau}\mu^{(i)}\|_2^2 \tag{27}$$

To compute $L$, the Lipschitz constant of $\nabla f$, we have

$$\begin{aligned}
\nabla f(x) &= 2\rho\left(A^T A(x - b) + \tau(x - c)\right) \\
&= 2(\rho A^T A + \tau I)x - 2(\rho A^T Ab + \tau c)
\end{aligned} \tag{28}$$

where $c = y^{(i)} - \frac{1}{\tau}\mu^{(i)}$. Thus, $L$ is just equal to the largest eigenvalue of $2(\rho A^T A + \tau I)$.

As recommended in Huang et al. (2016), ADMM provides flexibility to use various types of loss functions and regularizations without changing the procedure. For example, we can simply change to $L_{2,1}$ norm and equation 22 becomes a constrained ridge-regression problem, which can be efficiently solved by non-negative

quadratic programming algorithms. For most clinical usage, the size of questionnaire data is manageable on a single machine. However, if optimal computational and memory efficiency is required, various stochastic optimization approaches such as Mairal et al. (2010) can replace the ADMM procedure. Yet, an unbiased sampling scheme for generating random batches that handles missing responses is also needed. Such a scheme is non-trivial to obtain, especially under the multi-questionnaires scenario.

**Sub-problem 3 (Equation 3)**

Since both terms in equation 3 are in Frobenius-norm, $Z$ can be optimized entry-wise. In particular, we have the following closed-form solution for $Z^{(i+1)}$:

$$Z^{(i+1)} = \underset{[\min(M),\max(M)]}{Proj} \left( \mathcal{M} \odot M + \rho[W^{(i+1)}, C](Q^{(i+1)})^T - \alpha_Z^{(i)} \right) \oslash (\rho \mathbb{1} + \mathcal{M}) \tag{29}$$

where $\mathbb{1}$ is a 1-matrix with appropriate dimension and $\oslash$ is the Hadamard division.

## B    Optimization procedure using coordinate descent method

The subproblems for updating $W$, $Q$ and $Z$ employ the FISTA algorithm is efficient for large dataset. However, for small or moderate data sizes, coordinate descent algorithm outperforms FISTA. In the following, we revisit the Sub-problem 2 and work out all implementation details. Recall the sub-problem 2 that updates $W$:

$$W^{(i+1)} = \arg\min_{W \in \mathcal{W}} \frac{\rho}{2} \left\| Z^{(i)} - W \left(Q^{(i)}\right)^T - C \left(Q_C^{(i)}\right)^T + \frac{1}{\rho}\alpha_Z^{(i)} \right\|_F^2 + \beta_W \|W\|_{1,1} \tag{30}$$

To simplify the notation, we first divide the energy by $\frac{1}{\rho}$, followed by setting $\beta_W = 0$. We rewrite it into a more compact expression by defining

$$V := Z^{(i)} - C \left(Q_C^{(i)}\right)^T + \frac{1}{\rho}\alpha_Z^{(i)}, \quad H := \left(Q^{(i)}\right)^T, \quad W := W^{(i+1)} \tag{31}$$

Coordinate descent methods aim to conduct the following 1-variable update:

$$(W, H) \leftarrow (W + sE_{ir}, H) \tag{32}$$

where $E_{ir}$ is a matrix same size as $W$ with all entries 0 except the $(i, r)$ element which equals 1. This update is equivalent to solve the following 1-variable optimization problem to get $s$:

$$\min_{s:0 \leq W_{ir}+s \leq B_W} g_{ir}^W(s) \equiv \frac{1}{2} \left(V_{ij} - (WH)_{ij} - s \cdot H_{rj}\right)^2 \tag{33}$$

Simplifying $g_{ir}^W(s)$, we have

$$g_{ir}^W(s) = \frac{1}{2}\sum_j \left(V_{ij} - (WH)_{ij} - sH_{rj}\right)^2 = g_{ir}^W(0) + \left(g_{ir}^W\right)'(0) \cdot s + \frac{1}{2}\left(g_{ir}^W\right)''(0) \cdot s^2 \tag{34}$$

Define

$$G^W := \nabla_W g_{ir}^W(s) = WHH^T - VH^T, \quad G^H := \nabla_H g_{ir}^W(s) = W^TWH - W^TV \tag{35}$$

we then have

$$\left(g_{ir}^W\right)'(0) = \left(G^W\right)_{ir} = (WHH^T - VH^T)_{ir}, \quad \left(g_{ir}^W\right)''(0) = \left(HH^T\right)_{rr} \tag{36}$$

and the closed form solution of $s$ is:

$$s^* = \min\left(\max\left(0, W_{ir} - (WHH^T - VH^T)_{ir}/(HH^T)_{rr}\right), B_W\right) - W_{ir} \tag{37}$$

Coordinate descent algorithms then update $W_{ir}$ to $W_{ir} \leftarrow W_{ir} + s^*$. If $\beta_W \neq 0$, we have,

$$W_{ir} \leftarrow \min\left(\max\left(0, W_{ir} - \left((WHH^T - VH^T)_{ir} + \frac{\beta_W}{\rho}\right)\bigg/(HH^T)_{rr}\right), B_W\right) \tag{38}$$

Notice that the update of $W$ and $H$ is invariant to the missing entries of $M$ here as $Z$ is present as a surrogate of $M$. By taking transpose, the update of $H$ can be done similarly. These update are performed in cyclic on $W$, and followed by updates on $H$ for the two sub-problems.

## C    Details and proof of Preposition 3.1

In the following, we provide a self-contained convergence proof and show that, under an appropriate choice of the penalty parameter $\rho$, the ADMM optimization scheme discussed in Section 3.2 converges to a local minimum. To simplify notation, we denote $\mathbb{V}^{(i,j,k)} = \{W^{(i)}, Q^{(j)}, Z^{(k)}\}$ to be the tuple of variables $W, Q$ and $Z$ during iteration $(i), (j)$ and $(k)$ respectively. If $i = j = k$, we abbreviate it as $\mathbb{V}^{(i)}$. We also denote $R^{(i)} = [W^{(i)}, C](Q^{(i)})^T$ and for any matrices $A, B$ with appropriate dimensions, $\langle A, B \rangle = \text{Trace}(A^TB)$. In

the following, we are going to show that the Lagrangian is decreasing across iterations. Particularly, we consider the difference of Lagrangian between consecutive iterations:

$$
\begin{aligned}
&\mathcal{L}_\rho(\mathbb{V}^{(i+1)}, \alpha_Z^{(i+1)}) - \mathcal{L}_\rho(\mathbb{V}^{(i)}, \alpha_Z^{(i)}) \\
&= \underbrace{\mathcal{L}_\rho(\mathbb{V}^{(i+1)}, \alpha_Z^{(i+1)}) - \mathcal{L}_\rho(\mathbb{V}^{(i+1)}, \alpha_Z^{(i)})}_{(I)} + \underbrace{\mathcal{L}_\rho(\mathbb{V}^{(i+1)}, \alpha_Z^{(i)}) - \mathcal{L}_\rho(\mathbb{V}^{(i)}, \alpha_Z^{(i)})}_{(II)}
\end{aligned} \tag{39}
$$

Expanding term $(I)$, we have

$$
\mathcal{L}_\rho(\mathbb{V}^{(i+1)}, \alpha_Z^{(i+1)}) - \mathcal{L}_\rho(\mathbb{V}^{(i+1)}, \alpha_Z^{(i)}) = \left\langle \alpha_Z^{(i+1)} - \alpha_Z^{(i)}, Z^{(i+1)} - R^{(i+1)} \right\rangle = \frac{1}{\rho}\|\alpha_Z^{(i+1)} - \alpha_Z^{(i)}\|_F^2 \tag{40}
$$

Expanding term $(II)$, we have

$$
\mathcal{L}_\rho(\mathbb{V}^{(i+1)}, \alpha_Z^{(i)}) - \mathcal{L}_\rho(\mathbb{V}^{(i)}, \alpha_Z^{(i)}) \tag{41}
$$

$$
= \overbrace{\mathcal{L}_\rho(\mathbb{V}^{(i+1)}, \alpha_Z^{(i)}) - \mathcal{L}_\rho(\mathbb{V}^{(i+1,i+1,i)}, \alpha_Z^{(i)})}^{(\mathcal{A})} + \overbrace{\mathcal{L}_\rho(\mathbb{V}^{(i+1,i+1,i)}, \alpha_Z^{(i)}) - \mathcal{L}_\rho(\mathbb{V}^{(i+1,i,i)}, \alpha_Z^{(i)})}^{(\mathcal{B})}
$$

$$
+ \underbrace{\mathcal{L}_\rho(\mathbb{V}^{(i+1,i,i)}, \alpha_Z^{(i)}) - L(\mathbb{V}^{(k)}, \alpha_Z^{(i)})}_{(\mathcal{C})} \tag{42}
$$

Expanding $(\mathcal{A})$ by the definition, we have

$$
\begin{aligned}
&\frac{1}{2}\|\mathcal{M} \odot (M - Z^{(i+1)})\|_F^2 - \frac{1}{2}\|\mathcal{M} \odot (M - Z^{(i)})\|_F^2 + \left\langle \alpha_Z^{(i)}, Z^{(i+1)} - R^{(i+1)} \right\rangle \\
&\quad - \left\langle \alpha_Z^{(i)}, Z^{(i)} - R^{(i+1)} \right\rangle + \frac{\rho}{2}\left\| Z^{(i+1)} - R^{(i+1)} \right\|_F^2 - \frac{\rho}{2}\left\| Z^{(i)} - R^{(i+1)} \right\|_F^2 \\
&= \left\langle \mathcal{M} \odot (Z^{(i+1)} - M), \mathcal{M} \odot (Z^{(i+1)} - Z^{(i)}) \right\rangle - \|\mathcal{M} \odot (Z^{(i+1)} - Z^{(i)})\|_F^2 \\
&\quad + \left\langle \alpha_Z^{(i)}, Z^{(i+1)} - Z^{(i)} \right\rangle + \rho \left\langle Z^{(i+1)} - R^{(i+1)}, Z^{(i+1)} - Z^{(i)} \right\rangle - \rho\|Z^{(i+1)} - Z^{(i)}\|_F^2 \\
&= \left\langle \mathcal{M} \odot (Z^{(i+1)} - M) + \rho \cdot Z^{(i+1)} + \alpha_Z^{(i)} - \rho R^{(i+1)}, Z^{(i+1)} - Z^{(i)} \right\rangle \\
&\quad - \|\mathcal{M} \odot (Z^{(i+1)} - Z^{(i)})\|_F^2 - \rho\|(Z^{(i+1)} - Z^{(i)})\|_F^2 - \left\langle \mathcal{M} \odot (Z^{(i+1)} - M), (1 - \mathcal{M}) \odot (Z^{(i+1)} - Z^{(i)}) \right\rangle
\end{aligned}
$$

Since $Z^{(i+1)}$ is the minimizer of equation 3, we have

$$
\left\| \mathcal{M} \odot (M - Z^{(i+1)}) \right\|_F^2 + \rho \left\| Z^{(i+1)} - R^{(i+1)} + \frac{1}{\rho}\alpha_Z^{(i)} \right\|_F^2 \leq \left\| \mathcal{M} \odot (M - Z^{(i)}) \right\|_F^2 + \rho \left\| Z^{(i)} - R^{(i+1)} + \frac{1}{\rho}\alpha_Z^{(i)} \right\|_F^2
$$

which gives

$$
\begin{aligned}
&2 \left\langle \mathcal{M} \odot (Z^{(i+1)} - M), \mathcal{M} \odot (Z^{(i+1)} - Z^{(i)}) \right\rangle - \|\mathcal{M} \odot (Z^{(i+1)} - Z^{(i)})\|_F^2 \\
&\leq -2 \left\langle \rho \cdot Z^{(i+1)} + \alpha_Z^{(i)} - \rho R^{(i+1)}, Z^{(i+1)} - Z^{(i)} \right\rangle + \rho\|Z^{(i+1)} - Z^{(i)}\|_F^2
\end{aligned}
$$

It further implies

$$
\begin{aligned}
&\left\langle \rho \cdot Z^{(i+1)} + \alpha_Z^{(i)} - \rho R^{(i+1)}, Z^{(i+1)} - Z^{(i)} \right\rangle \\
&\leq - \left\langle \mathcal{M} \odot (Z^{(i+1)} - M), \mathcal{M} \odot (Z^{(i+1)} - Z^{(i)}) \right\rangle + \frac{1}{2}\|\mathcal{M} \odot (Z^{(i+1)} - Z^{(i)})\|_F^2 + \frac{\rho}{2}\|Z^{(i+1)} - Z^{(i)}\|_F^2
\end{aligned}
$$

By direct substitution, we have

$$\begin{aligned}
(\mathcal{A}) &\leq \left\langle \mathcal{M} \odot (Z^{(i+1)} - M), Z^{(i+1)} - Z^{(i)} \right\rangle - \left\langle \mathcal{M} \odot (Z^{(i+1)} - M), \mathcal{M} \odot (Z^{(i+1)} - Z^{(i)}) \right\rangle \\
&\quad + \frac{1}{2}\|\mathcal{M} \odot (Z^{(i+1)} - Z^{(i)}\|_F^2 + \frac{\rho}{2}\|Z^{(i+1)} - Z^{(i)}\|_F^2 - \|\mathcal{M} \odot (Z^{(i+1)} - Z^{(i)})\|_F^2 \\
&\quad - \rho\|(Z^{(i+1)} - Z^{(i)})\|_F^2 - \left\langle \mathcal{M} \odot (Z^{(i+1)} - M), (1 - \mathcal{M}) \odot (Z^{(i+1)} - Z^{(i)}) \right\rangle \\
&= -\frac{1}{2}\|\mathcal{M} \odot (Z^{(i+1)} - Z^{(i)})\|_F^2 - \frac{\rho}{2}\|(Z^{(i+1)} - Z^{(i)})\|_F^2 \leq -\frac{\rho}{2}\|(Z^{(i+1)} - Z^{(i)})\|_F^2
\end{aligned} \tag{43}$$

For the second term $(\mathcal{B})$, by definition, we have,

$$\begin{aligned}
(\mathcal{B}) &= \frac{\rho}{2}\left\| Z^{(i)} - R^{(i+1)} + \frac{1}{\rho}\alpha_Z^{(i)} \right\|_F^2 - \frac{\rho}{2}\left\| Z^{(i)} - [W^{(i+1)}, C]Q^{(i),T} + \frac{1}{\rho}\alpha_Z^{(i)} \right\|_F^2 + \beta\|Q^{(i+1)}\|_{1,1} - \beta\|Q^{(i)}\|_{1,1} \\
&= \rho\left\langle R^{(i+1)} - Z^{(i)} - \frac{1}{\rho}\alpha_Z^{(i)}, [W^{(i+1)}, C](Q^{(i+1),T} - Q^{(i),T}) \right\rangle \\
&\quad - \frac{\rho}{2}\left\| [W^{(i+1)}, C](Q^{(i+1),T} - Q^{(i),T}) \right\|_F^2 + \beta(\|Q^{(i+1)}\|_{1,1} - \|Q^{(i)}\|_{1,1})
\end{aligned}$$

We recall that $Q$ is updated via solving constrained Lasso problems for every row $Q_{[r,:]}^{(i+1)}$:

$$y = \underset{x, 0 \leq x}{\arg\min}\ \beta\|x\|_1 + \frac{\rho}{2}\|b - Ax\|_2^2, \quad \text{where} \quad A = [W^{(i+1)}, C], \quad b = \left[ Z^{(i)} + \frac{1}{\rho}\alpha_Z^{(i)} \right]_{[r,:]}$$

One obtains $y$ if and only if there exists $g \in \partial\|y\|_1$, the sub-differential of $\|\cdot\|_1$ such that

$$\rho A^T(Ay - b) + \beta g = \mathbf{0}. \tag{44}$$

As $\|\cdot\|_1$ is convex, we have

$$\|x\|_1 \geq \|y\|_1 + \langle x - y, g\rangle \tag{45}$$

which gives

$$\|y\|_1 - \|x\|_1 \leq \left\langle y - x, \frac{\rho}{\beta}A^T(Ay - b) \right\rangle = \left\langle A(y - x), \frac{\rho}{\beta}(Ay - b) \right\rangle \tag{46}$$

Re-substituting $x = Q_{[r,:]}^{(i),T}$, $y = Q_{[r,:]}^{(i+1),T}$, $A = [W^{(i+1)}, C]$, $b = \left[ Z^{(i)} + \frac{1}{\rho}\alpha_Z^{(i)} \right]_{[r,:]}$ and sum over $r$, we have

$$\beta\|Q^{(i+1)}\|_{1,1} - \beta\|Q^{(i)}\|_{1,1} \leq -\rho\left\langle R^{(i+1)} - Z^{(i)} - \frac{1}{\rho}\alpha_Z^{(i)}, [W^{(i+1)}, C](Q^{(i+1),T} - Q^{(i),T}) \right\rangle \tag{47}$$

Therefore, we have

$$(\mathcal{B}) \leq -\frac{\rho}{2}\left\| [W^{(i+1)}, C](Q^{(i+1),T} - Q^{(i),T}) \right\|_F^2 \tag{48}$$

With similar argument, we can bound $(\mathcal{C})$ by

$$(\mathcal{C}) \leq -\frac{\rho}{2}\left\| [(W^{(i+1)} - W^{(i)}), C]Q^{(i),T} \right\|_F^2 \tag{49}$$

To get an upper bound of $\|\alpha_{\boldsymbol{Z}}^{(i+1)} - \alpha_{\boldsymbol{Z}}^{(i)}\|_F^2$, we have

$$\begin{aligned}
&\|\alpha_{\boldsymbol{Z}}^{(i+1)} - \alpha_{\boldsymbol{Z}}^{(i)}\|_F^2 \\
&\leq \|\boldsymbol{Z}^{(i+1)} - \boldsymbol{Z}^{(i)}\|_F^2 + \|\boldsymbol{R}^{(i+1)} - \boldsymbol{R}^{(i)}\|_F^2 \\
&\leq \|\boldsymbol{Z}^{(i+1)} - \boldsymbol{Z}^{(i)}\|_F^2 + \|[\boldsymbol{W}^{(i+1)}, \boldsymbol{C}]\boldsymbol{Q}^{(i+1),T} - [\boldsymbol{W}^{(i+1)}, \boldsymbol{C}]\boldsymbol{Q}^{(i),T}\|_F^2 \\
&\quad + \|[\boldsymbol{W}^{(i+1)}, \boldsymbol{C}]\boldsymbol{Q}^{(i),T} - [\boldsymbol{W}^{(i)}, \boldsymbol{C}]\boldsymbol{Q}^{(i),T}\|_F^2 \\
&\leq \|\boldsymbol{Z}^{(i+1)} - \boldsymbol{Z}^{(i)}\|_F^2 + \|[\boldsymbol{W}^{(i+1)}, \boldsymbol{C}](\boldsymbol{Q}^{(i+1),T} - \boldsymbol{Q}^{(i),T})\|_F^2 + \|[(\boldsymbol{W}^{(i+1)} - \boldsymbol{W}^{(i)}), \boldsymbol{C}]\boldsymbol{Q}^{(i),T}\|_F^2
\end{aligned} \tag{50}$$

Combining equation 40, 50, 42, 43, 48 and 49 with equation 39, we have

$$
\begin{aligned}
&\mathcal{L}_\rho(\mathbb{V}^{(i+1)}, \alpha_{\boldsymbol{Z}}^{(i+1)}) - \mathcal{L}_\rho(\mathbb{V}^{(i)}, \alpha_{\boldsymbol{Z}}^{(i)}) \\
&\leq \frac{1}{\rho} \left\| \alpha_{\boldsymbol{Z}}^{(i+1)} - \alpha_{\boldsymbol{Z}}^{(i)} \right\|_F^2 - \frac{\rho}{2} \left\| \boldsymbol{Z}^{(i+1)} - \boldsymbol{Z}^{(i)} \right\|_F^2 - \frac{\rho}{2} \left\| [\boldsymbol{W}^{(i+1)}, \boldsymbol{C}](\boldsymbol{Q}^{(i+1),T} - \boldsymbol{Q}^{(i),T}) \right\|_F^2 \\
&\quad - \frac{\rho}{2} \left\| [(\boldsymbol{W}^{(i+1)} - \boldsymbol{W}^{(i)}), \boldsymbol{C}]\boldsymbol{Q}^{(i),T} \right\|_F^2 \\
&\leq \left( \frac{1}{\rho} - \frac{\rho}{2} \right) \cdot \left( \|\boldsymbol{Z}^{(i+1)} - \boldsymbol{Z}^{(i)}\|_F^2 + \|[\boldsymbol{W}^{(i+1)}, \boldsymbol{C}](\boldsymbol{Q}^{(i+1),T} - \boldsymbol{Q}^{(i),T})\|_F^2 \right. \\
&\quad \left. + \|[(\boldsymbol{W}^{(i+1)} - \boldsymbol{W}^{(i)}), \boldsymbol{C}]\boldsymbol{Q}^{(i),T}\|_F^2 \right)
\end{aligned}
\tag{51}
$$

This is summarized into the following theorem:

*Theorem* C.1 (Non-increasing property). Assume $\rho \geq \sqrt{2}$, for all $i$, we have

$$
\mathcal{L}_\rho(\boldsymbol{W}^{(i+1)}, \boldsymbol{Q}^{(i+1)}, \boldsymbol{Z}^{(i+1)}, \alpha_{\boldsymbol{Z}}^{(i+1)}) \leq \mathcal{L}_\rho(\boldsymbol{W}^{(i)}, \boldsymbol{Q}^{(i)}, \boldsymbol{Z}^{(i)}, \alpha_{\boldsymbol{Z}}^{(i)}).
\tag{52}
$$

We set $\rho = 3$ in all experiments for sufficiency.

# D   Details and proof of Proposition 3.2

Assume that there is a ground-truth factorization $(\mathbf{W}^*, \mathbf{Q}^*)$ of the given $\mathbf{M} = \mathbf{W}^*(\mathbf{Q}^*)^T$, with latent dimension $k^*$, where $\mathbf{W}^*$ and $\mathbf{Q}^*$ are matrix-valued random variables with entries sampled from some bounded distributions. With high probability, the error $\|\mathbf{M} - \mathbf{W}\mathbf{Q}^T\|_F^2$ we are minimizing is star-convex towards $(\mathbf{W}^*, \mathbf{Q}^*)$ whenever $k = k^*$ (Bjorck et al., 2021). To demonstrate the importance of the choice of $k$, we consider the scenario when $k \neq k^*$ below.

First, a more precise assumption for ICQF is to model $\mathbf{W}$ as *row-independent bounded random matrices*. Recall that $\boldsymbol{W}$ is generated by arranging $n$ participants' latent representation as rows of $n \times k$ matrix, where the $n$ participants are assumed to be independent from each other and their corresponding latent representations follow a high-dimensional bounded distribution.

Second, let $(\mathbf{W}_1, \mathbf{Q}_1)$ and $(\mathbf{W}_2, \mathbf{Q}_2)$ be two factorizations with dimensions $k_1$ and $k_2$ respectively. Assume both factorizations achieve **(a)**: equivalent mismatching loss in expectation, and **(b)**: equivalent expectation approximation to data matrix $\mathbf{M}$:

$$
\textbf{(a)}: \ \mathbb{E}\left[\|\mathbf{M} - \mathbf{W}_1\mathbf{Q}_1^T\|_F^2\right] = \mathbb{E}\left[\|\mathbf{M} - \mathbf{W}_2\mathbf{Q}_2^T\|_F^2\right] \quad \text{and} \quad \textbf{(b)}: \ \mathbb{E}[\mathbf{W}_1\mathbf{Q}_1^T] = \mathbb{E}[\mathbf{W}_2\mathbf{Q}_2^T]
$$

We also assume **(c)**: $\mathbb{E}\left[\sum_{j=1}^n (\mathbf{W}_i)_{j\kappa}^2\right] := \sigma_{\mathbf{W}_i}^2$ and $\mathbb{E}\left[\sum_{j=1}^m (\mathbf{Q}_i)_{j\kappa}^2\right] := \sigma_{\mathbf{Q}_i}^2$ for all $\kappa = k_i$, $i = 1, 2$.

Expanding **(a)**, we have

$$
\mathbb{E}\left[\text{Trace}\left((\mathbf{M} - \mathbf{W}_1\mathbf{Q}_1^T)^T(\mathbf{M} - \mathbf{W}_1\mathbf{Q}_1^T)\right)\right] = \mathbb{E}\left[\text{Trace}\left((\mathbf{M} - \mathbf{W}_2\mathbf{Q}_2^T)^T(\mathbf{M} - \mathbf{W}_2\mathbf{Q}_2^T)\right)\right]
$$

This gives

$$
\mathbb{E}\left[\text{Trace}\left(\mathbf{W}_1^T\mathbf{W}_1\mathbf{Q}_1^T\mathbf{Q}_1 - 2\mathbf{M}^T\mathbf{W}_1\mathbf{Q}_1^T\right)\right] = \mathbb{E}\left[\text{Trace}\left(\mathbf{W}_2^T\mathbf{W}_2\mathbf{Q}_2^T\mathbf{Q}_2 - 2\mathbf{M}^T\mathbf{W}_2\mathbf{Q}_2^T\right)\right]
$$

Denote $\mathbb{E}[\mathbf{W}_i] = \mu_{\mathbf{W}_i}$, $\mathbb{E}[\mathbf{Q}_i] = \mu_{\mathbf{Q}_i}$ for $i = 1, 2$, we have $\mathbf{W}_i = \bar{\mathbf{W}}_i + \mu_{\mathbf{W}_i}$ and $\mathbf{Q}_i = \bar{\mathbf{Q}}_i + \mu_{\mathbf{Q}_i}$, where $\bar{\mathbf{W}}_i$ and $\bar{\mathbf{Q}}_i$ denote the corresponding centered variables. Note that by the independence of $\mathbf{W}_i$ and $\mathbf{Q}_i$ and linearity of trace and expectation operator,

$$
\begin{aligned}
&\mathbb{E}\left[\text{Trace}\left(\mathbf{M}^T\mathbf{W}_1\mathbf{Q}_1^T\right)\right] \\
&= \mathbb{E}\left[\text{Trace}\left(\mathbf{M}^T\bar{\mathbf{W}}_1\bar{\mathbf{Q}}_1^T + \mathbf{M}^T\bar{\mathbf{W}}_1\mu_{\mathbf{Q}_1}^T + \mathbf{M}^T\mu_{\mathbf{W}_1}\bar{\mathbf{Q}}_1^T + \mathbf{M}^T\mu_{\mathbf{W}_1}\mu_{\mathbf{Q}_1}^T\right)\right] \\
&= \text{Trace}(\mathbf{M}^T\mathbb{E}[\mathbf{W}_1]\mathbb{E}[\mathbf{Q}_1^T]) = \text{Trace}(\mathbf{M}^T\mathbb{E}[\mathbf{W}_2]\mathbb{E}[\mathbf{Q}_2^T]) = \mathbb{E}\left[\text{Trace}\left(\mathbf{M}^T\mathbf{W}_2\mathbf{Q}_2^T\right)\right]
\end{aligned}
\tag{53}
$$

which yields

$$\mathbb{E}\left[\text{Trace}\left(\mathbf{W}_1^T\mathbf{W}_1\mathbf{Q}_1^T\mathbf{Q}_1\right)\right] = \mathbb{E}\left[\text{Trace}\left(\mathbf{W}_2^T\mathbf{W}_2\mathbf{Q}_2^T\mathbf{Q}_2\right)\right] \tag{54}$$

Consider $\mathbb{E}\left[\text{Trace}\left(\mathbf{W}_1^T\mathbf{W}_1\mathbf{Q}_1^T\mathbf{Q}_1\right)\right]$ via definition, we have

$$\mathbb{E}\left[\text{Trace}\left(\mathbf{W}_1^T\mathbf{W}_1\mathbf{Q}_1^T\mathbf{Q}_1\right)\right]$$
$$=\text{Trace}\left(\mathbb{E}\left[\mathbf{W}_1^T\mathbf{W}_1\right]\mathbb{E}\left[\mathbf{Q}_1^T\mathbf{Q}_1\right]\right)$$
$$=\text{Trace}\left(\mathbb{E}\begin{bmatrix}\left(\sum_{j=1}^n(\mathbf{W}_1)_{j1}^2\right) & & * \\ & \ddots & \\ * & & \left(\sum_{j=1}^n(\mathbf{W}_1)_{jk_1}^2\right)\end{bmatrix} \times \mathbb{E}\begin{bmatrix}\left(\sum_{j=1}^m(\mathbf{Q}_1)_{j1}^2\right) & & 0 \\ & \ddots & \\ 0 & & \left(\sum_{j=1}^m(\mathbf{Q}_1)_{jk_1}^2\right)\end{bmatrix}\right)$$
$$=\sum_{\kappa=1}^{k_1}\mathbb{E}\left[\sum_{j=1}^n(\mathbf{W}_1)_{j\kappa}^2\right]\mathbb{E}\left[\sum_{j=1}^m(\mathbf{Q}_1)_{j\kappa}^2\right] \tag{55}$$

Incorporating assumption **(c)**, we have

$$\mathbb{E}\left[\text{Trace}\left(\mathbf{W}_1^T\mathbf{W}_1\mathbf{Q}_1^T\mathbf{Q}_1\right)\right] = k_1\sigma_{\mathbf{W}_1}^2\sigma_{\mathbf{Q}_1}^2 \tag{56}$$

Consider equation 54 with $k_1 > k_2$. For $\mathbf{W}_1, \mathbf{Q}_1$, W.L.O.G. we pad $k_2 - k_1$ columns of zeros. Moreover, let $\mathbf{P}$ be an optimal $k_2 \times k_2$ permutation matrix, we also have

$$\mathbb{E}\left[\text{Trace}\left((\mathbf{W}_2\mathbf{P})^T\mathbf{W}_2\mathbf{P}(\mathbf{Q}_2\mathbf{P})^T\mathbf{Q}_2\mathbf{P}\right)\right] = \mathbb{E}\left[\text{Trace}\left(\mathbf{W}_2^T\mathbf{W}_2\mathbf{Q}_2^T\mathbf{Q}_2\right)\right] = k_2\sigma_{\mathbf{W}_2}^2\sigma_{\mathbf{Q}_2}^2 \tag{57}$$

Combining with equation 54, it is equivalent to

$$k_1\sigma_{\mathbf{W}_1}^2\sigma_{\mathbf{Q}_1}^2 = k_2\sigma_{\mathbf{W}_2}^2\sigma_{\mathbf{Q}_2}^2 \tag{58}$$

which gives

$$\mathbb{E}\left[\|\mathbf{W}_1\|_F^2\right] = \frac{\sigma_{\mathbf{Q}_2}^2}{\sigma_{\mathbf{Q}_1}^2}\mathbb{E}\left[\|\mathbf{W}_2\|_F^2\right] = \frac{\sigma_{\mathbf{Q}_2}^2}{\sigma_{\mathbf{Q}_1}^2}\mathbb{E}\left[\|\mathbf{W}_2\mathbf{P}\|_F^2\right] \tag{59}$$

To evaluate the impact of interpretability of latent representation under different latent dimension, we consider $\mathbb{E}\left[\|\mathbf{W}_1 - \mathbf{W}_2\mathbf{P}\|_F^2\right]$:

$$\mathbb{E}\left[\|\mathbf{W}_1 - \mathbf{W}_2\mathbf{P}\|_F^2\right] = \mathbb{E}\left[\text{Trace}\left((\mathbf{W}_1 - \mathbf{W}_2\mathbf{P})^T(\mathbf{W}_1 - \mathbf{W}_2\mathbf{P})\right)\right]$$
$$= \mathbb{E}\left[\|\mathbf{W}_1\|_F^2\right] + \frac{\sigma_{\mathbf{Q}_1}^2}{\sigma_{\mathbf{Q}_2}^2}\mathbb{E}\left[\|\mathbf{W}_1\|_F^2\right] - 2\mathbb{E}\left[\text{Trace}(\mathbf{W}_1^T\mathbf{W}_2\mathbf{P})\right] \tag{60}$$

As $\text{Trace}(\mathbf{W}_1^T\mathbf{W}_2 P) \leq \|\mathbf{W}_1\|_F\|\mathbf{W}_2\mathbf{P}\|_F$, we also have

$$\mathbb{E}\left[\text{Trace}(\mathbf{W}_1^T\mathbf{W}_2\mathbf{P})\right] \leq \mathbb{E}\left[\|\mathbf{W}_1\|_F\right]\cdot\mathbb{E}\left[\|\mathbf{W}_2\mathbf{P}\|_F\right]$$
$$\leq \sqrt{\mathbb{E}\left[\|\mathbf{W}_1\|_F^2\right]\cdot\mathbb{E}\left[\|\mathbf{W}_2\|_F^2\right]} = \sqrt{\frac{\sigma_{\mathbf{Q}_1}^2}{\sigma_{\mathbf{Q}_2}^2}}\mathbb{E}\left[\|\mathbf{W}_1\|_F^2\right] \tag{61}$$

which implies

$$\mathbb{E}\left[\|\mathbf{W}_1 - \mathbf{W}_2\mathbf{P}\|_F^2\right] \geq \left(1 - 2\sqrt{\frac{\sigma_{\mathbf{Q}_1}^2}{\sigma_{\mathbf{Q}_2}^2}} + \frac{\sigma_{\mathbf{Q}_1}^2}{\sigma_{\mathbf{Q}_2}^2}\right)\mathbb{E}\left[\|\mathbf{W}_1\|_F^2\right] = \left(1 - \sqrt{\frac{\sigma_{\mathbf{Q}_1}^2}{\sigma_{\mathbf{Q}_2}^2}}\right)^2\mathbb{E}\left[\|\mathbf{W}_1\|_F^2\right] \tag{62}$$

Since $\mathbf{W}_i$ is generated from row-wise independent bounded distribution, if we add a mild assumption that $\sigma_{\mathbf{W}_i}^2 := \sigma_{\mathbf{W}}^2$ for all $i$ through re-scaling, Equation 58 implies $k_1\sigma_{\mathbf{Q}_1}^2 = k_2\sigma_{\mathbf{Q}_2}^2$ and therefore

$$\mathbb{E}\left[\|\mathbf{W}_1 - \mathbf{W}_2\|_F^2\right] \geq \left(1 - 2\sqrt{\frac{k_2}{k_1}} + \frac{k_2}{k_1}\right)\mathbb{E}\left[\|\mathbf{W}_1\|_F^2\right] = \left(\sqrt{\frac{k_2}{k_1}} - 1\right)^2\mathbb{E}\left[\|\mathbf{W}_1\|_F^2\right] \tag{63}$$

If we substitute $k_1 = k^*$, $(\mathbf{W}_1, \mathbf{Q}_1) = (\mathbf{W}^*, \mathbf{Q}^*)$, we have

$$\mathbb{E}\left[\|\mathbf{W}^* - \mathbf{W}_2\|_F^2\right] \geq \left(\sqrt{\frac{k_2}{k^*}} - 1\right)^2 \mathbb{E}\left[\|\mathbf{W}^*\|_F^2\right] \tag{64}$$

which means the relative expected difference between $\mathbf{W}^*$ and $\mathbf{W}_2$ is bounded below by $\left(\sqrt{\frac{k_2}{k^*}} - 1\right)^2$.

To prove that equation 64 holds in general, we consider the matrix concentration inequalities and show that large deviations from their means are exponentially unlikely. Benefitting from the model constraints, we can further assume that $\boldsymbol{W}$ is generated from some high dimensional bounded distribution. In the following, we make use of the main theorem proposed in Meckes & Szarek (2012) on concentration of non-commutative random matrices polynomials. As $\mathbf{W}_i$ are generated from bounded distributions, $\|\mathbf{W}_i - \mathbb{E}[\mathbf{W}_i]\|_F$ is uniformly bounded. Therefore, it satisfies the convex concentration properties. The theorem achieves the following results:

$$\mathbb{P}\left\{\|\mathbf{W}\|_F^2 - \mathbb{E}\left[\|\mathbf{W}\|_F^2\right] > tkn^2\right\} \leq C_1 \exp\left(-C_2 \min(t^2, t^{1/2})n\right) \tag{65}$$

Recall that $\mathbb{E}\left[\|\mathbf{W}_1 - \mathbf{W}_2 P\|_F^2\right] = \mathbb{E}\left[\|\mathbf{W}_1\|_F^2\right] + \frac{\sigma_{\mathbf{Q}_1}^2}{\sigma_{\mathbf{Q}_2}^2}\mathbb{E}\left[\|\mathbf{W}_1\|_F^2\right] - 2\mathbb{E}\left[\text{Trace}(\mathbf{W}_1^T \mathbf{W}_2 \mathbf{P})\right]$. By padding $\mathbf{W}_1$ and $\mathbf{W}_2$ with zeros columns, we assume that $\mathbf{W}_i$ are all $n \times n$ matrices. Then the probability that the any one of the terms is deviating from their mean by a relative factor $\epsilon$ is less than $C_1 \exp(-C_2 \epsilon^2 n)$ for some small $\epsilon$. By the union bound, the probability that the either of them does is less than or equal to $C_3 \exp(-C_4 \epsilon^2 n)$.

# E   Visualization of the experimental setup for diagnostic prediction evaluation

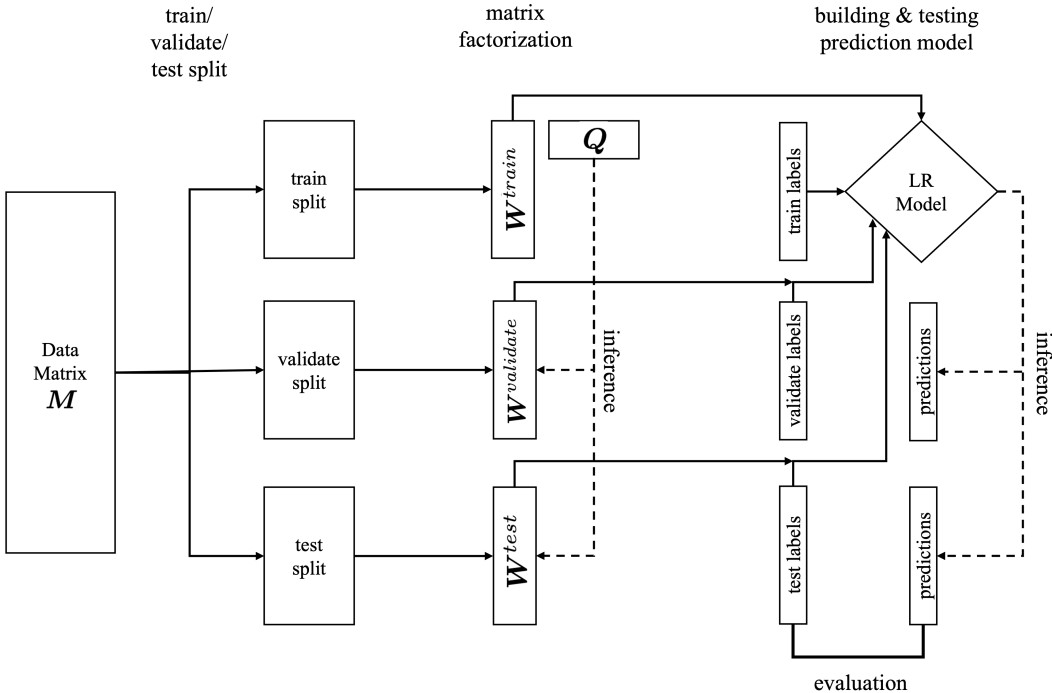

Figure 4: Setup for diagnostic prediction experiments.

## F   Table of the 21 questionnaires used in $HBN$ dataset

Table 3: Optimal $(k, \beta)$ of all 21 questionnaires.

| Questionnaire | Abbreviation | $n$ questions | Subscales | $k$ | $\beta$ |
|---|---|---|---|---|---|
| Affective Reactivity Index (Parent-Report) | ARI_P | 7 | nan | 2 | 0.01 |
| Affective Reactivity Index (Self-Report) | ARI_S | 7 | nan | 2 | 0.01 |
| Autism Spectrum Screening Questionnaire | ASSQ | 27 | nan | 2 | 0.01 |
| Conners 3 (Self-Report) | C3SR | 9 | | 4 | 0.05 |
| Child Behavior Checklist | CBCL | 119 | 9 | 8 | 0.5 |
| Extended Strengths and Weaknesses Assessment of Normal Behavior | ESWAN | 65 | nan | 13 | 0.2 |
| Inventory of Callous-Unemotional Traits (Parent-Report) | ICU_P | 24 | 3 | 4 | 0.1 |
| Inventory of Callous-Unemotional Traits (Self-Report) | ICU_SR | 24 | 3 | 3 | 0.1 |
| Mood and Feelings Questionnaire (Parent-Report) | MFQ_P | 34 | nan | 2 | 0.1 |
| Mood and Feelings Questionnaire (Self-Report) | MFQ_SR | 33 | nan | 2 | 0.1 |
| The Positive and Negative Affect Schedule | PANAS | 20 | 2 | 2 | 0.05 |
| Repetitive Behaviors Scale | RBS | 43 | 5 | 3 | 0.1 |
| Screen for Child Anxiety Related Disorders (Parent-Report) | SCARED_P | 41 | 5 | 3 | 0.1 |
| Screen for Child Anxiety Related Disorders (Self-Report) | SCARED_SR | 41 | 5 | 3 | 0.3 |
| Social Communication Questionnaire | SCQ | 40 | nan | 4 | 0.02 |
| Strength and Difficulties Questionnaire | SDQ | 33 | 9 | 6 | 0.05 |
| Social Responsiveness Scale (School Age) | SRS | 65 | 7 | 3 | 0.5 |
| The Strengths and Weaknesses Assessment of Normal Behavior Rating Scale for ADHD | SWAN | 18 | 2 | 3 | 0.02 |
| Symptom Checklist (Parent-Report) | SympChck | 63 | nan | 3 | 0.1 |
| Teacher Report Form (School Age) | TRF | 116 | 19 | 8 | 0.5 |
| Youth Self Report | YSR | 119 | 11 | 3 | 0.2 |

## G    Synthetic example and the factorization results

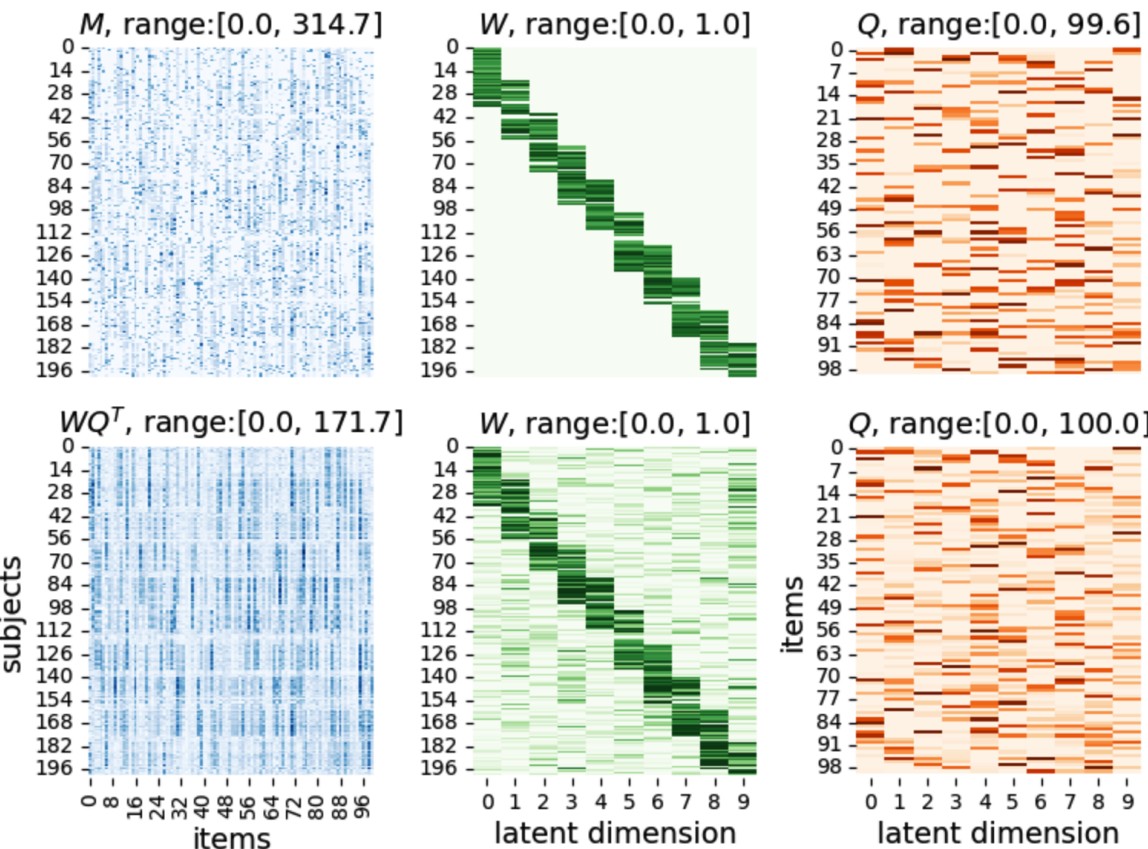

Figure 5: Top row: (left) The synthetic noisy data matrix $M$; (center): The ground-truth $W$; (right): The groud-truth $Q$. Bottom row: (left) Reconstructed $M = WQ^T$ based on the estimated $W$ and $Q$; (center): The estimated $W$ by ICQF; (right): The estimated $Q$ by ICQF.

## H Question embedding ($Q$) obtained from the Factor analysis, and the proposed method with optimal latent dimension.

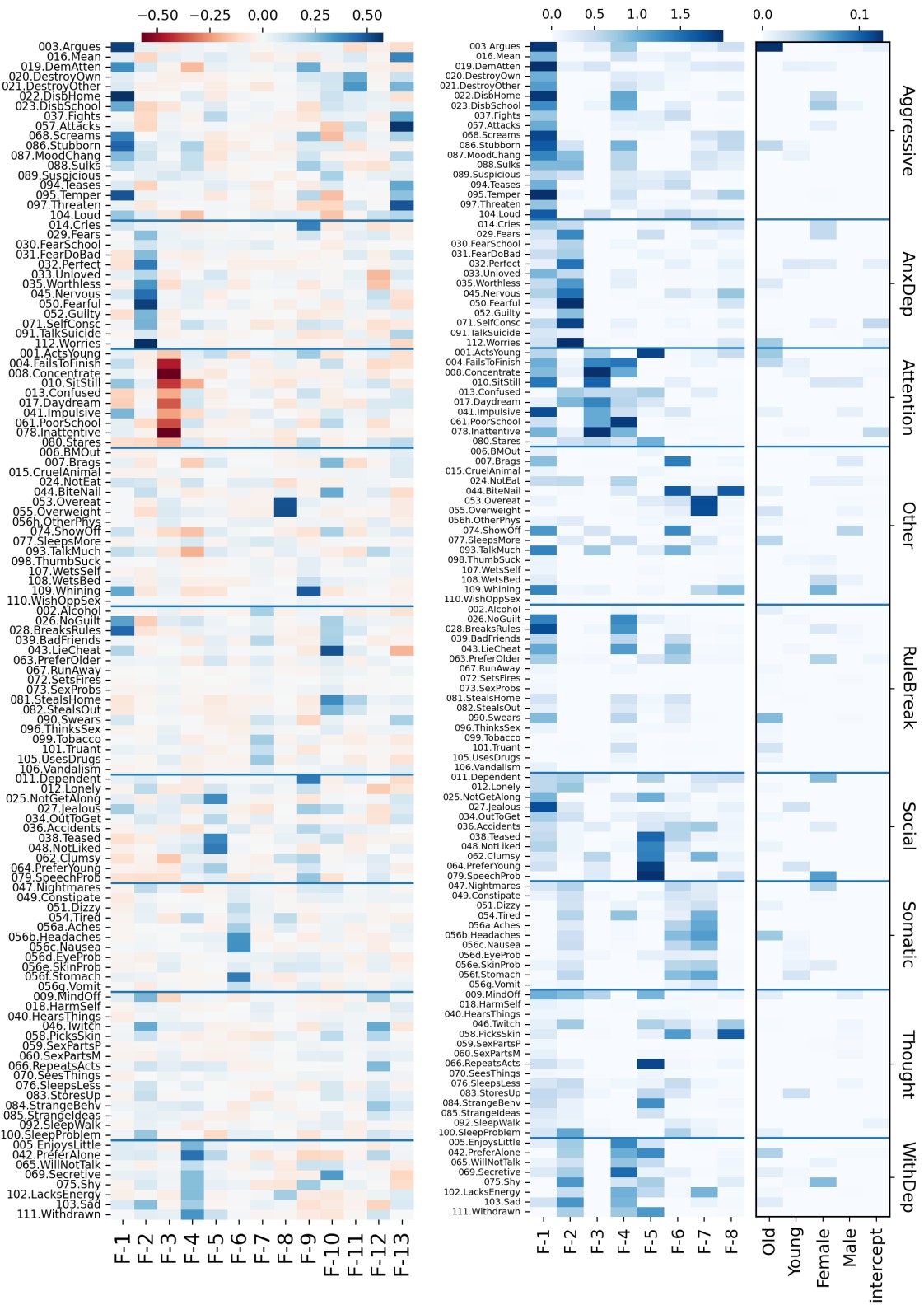

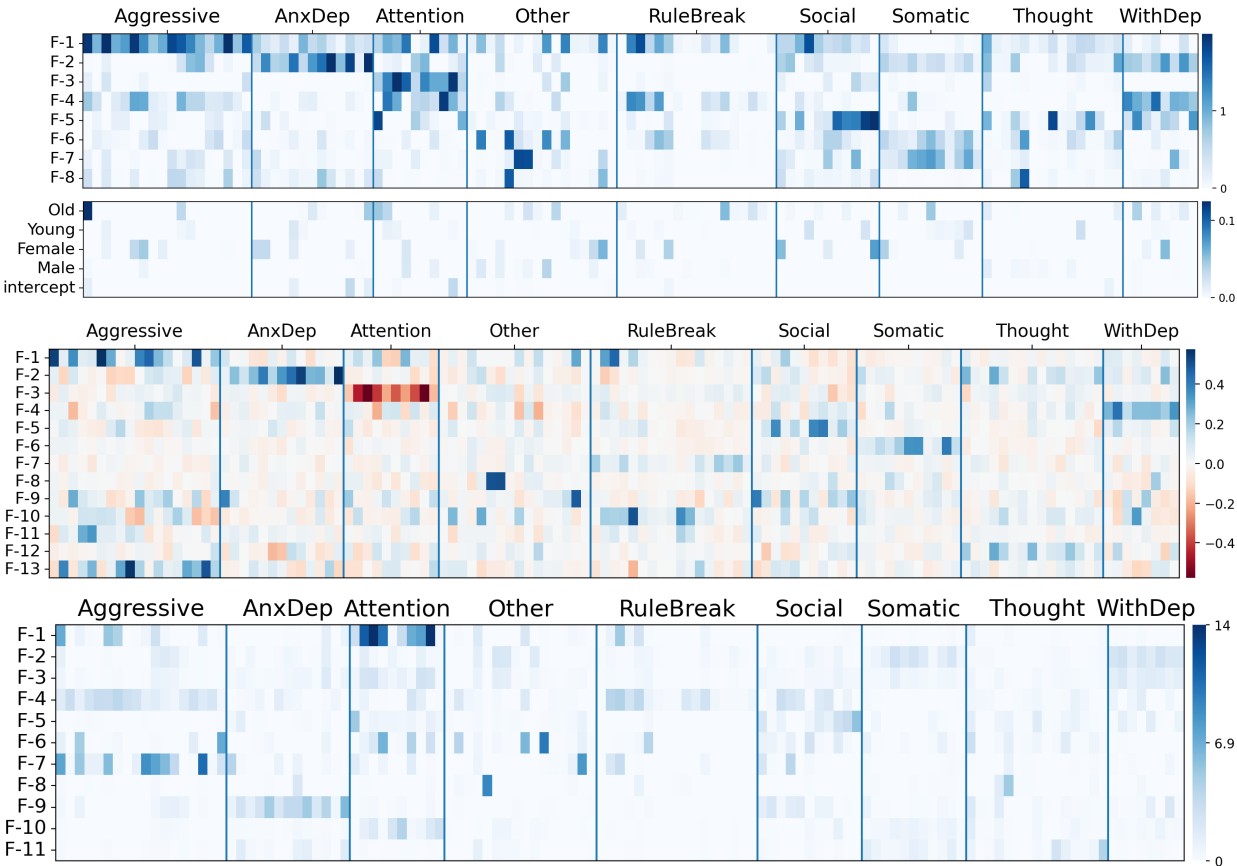

Figure 6: Horizontal heatmap of factor loadings $Q := [^R Q, ^C Q]$ from ICQF (top), Factor Analysis with promax rotation (middle) and $\ell_1$-NMF (bottom).

# I   Visualization and subjective evaluation of question embedding $Q$ in CBCL-HBN questionnaire

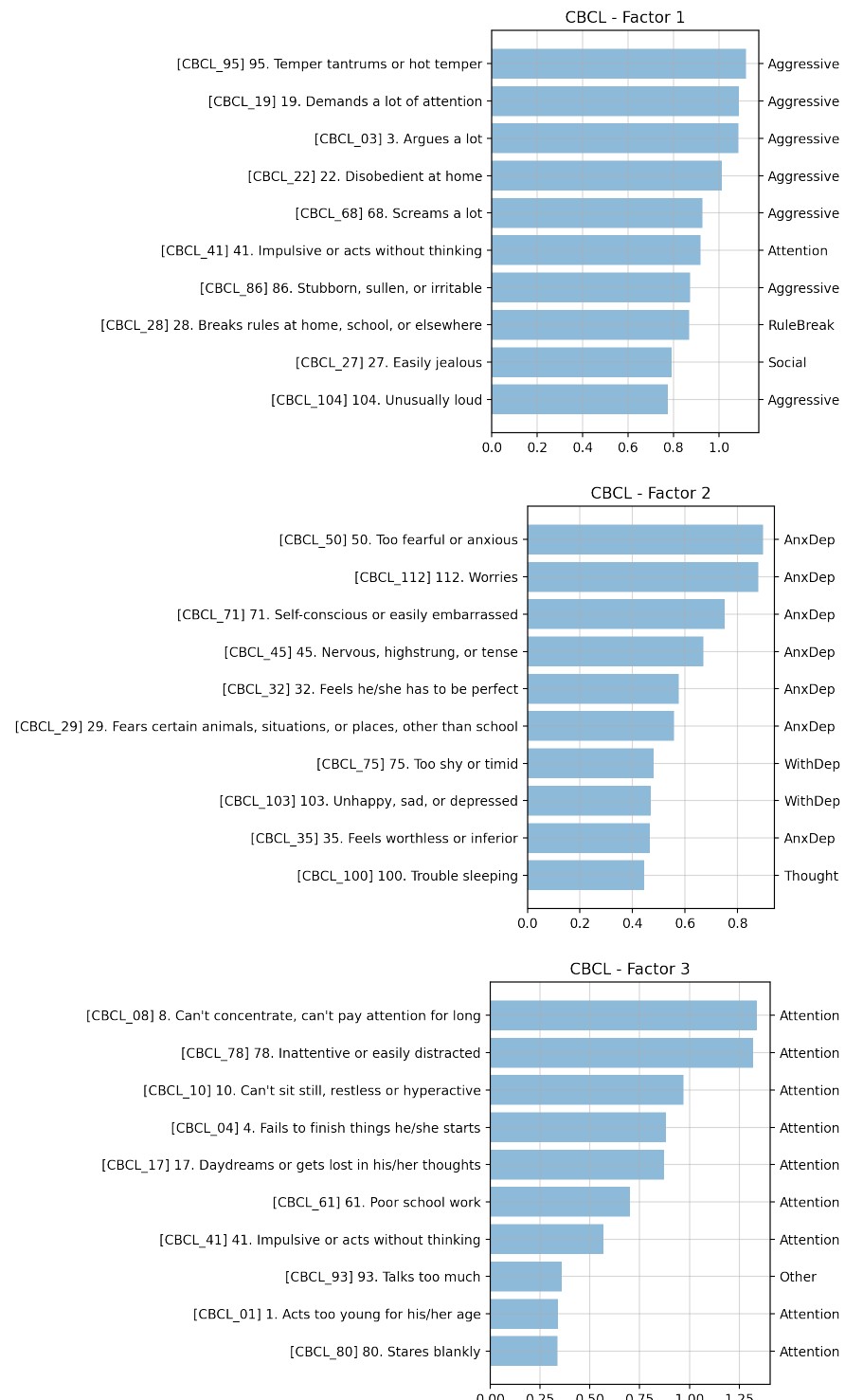

Figure 7: Top 10 questions ranked by $Q$ in *CBCL* using $Q$ obtained from ICQF (Factor 1-3).

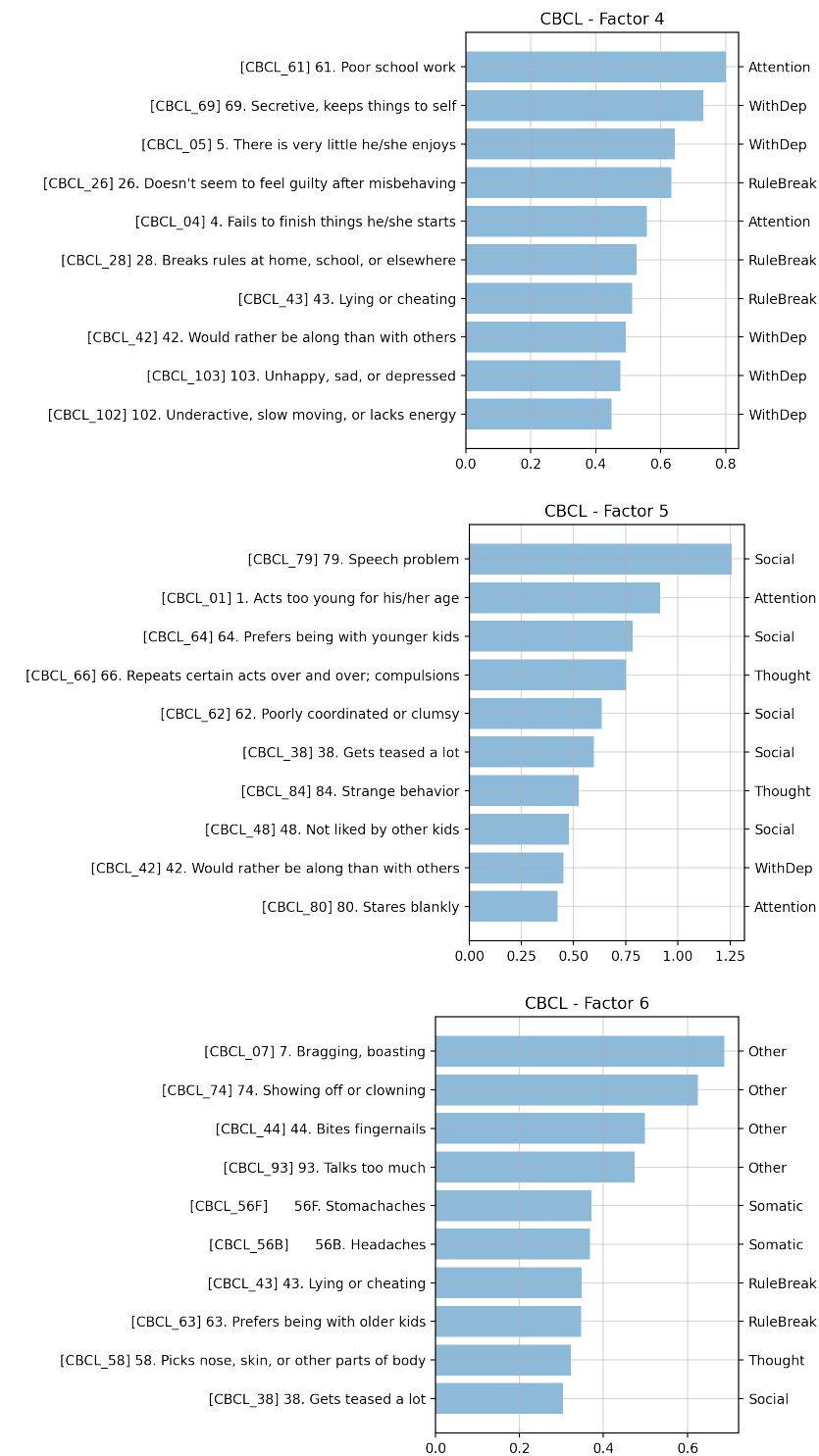

Figure 8: Top 10 questions ranked by $Q$ in *CBCL* using $Q$ obtained from ICQF (Factor 4-6).

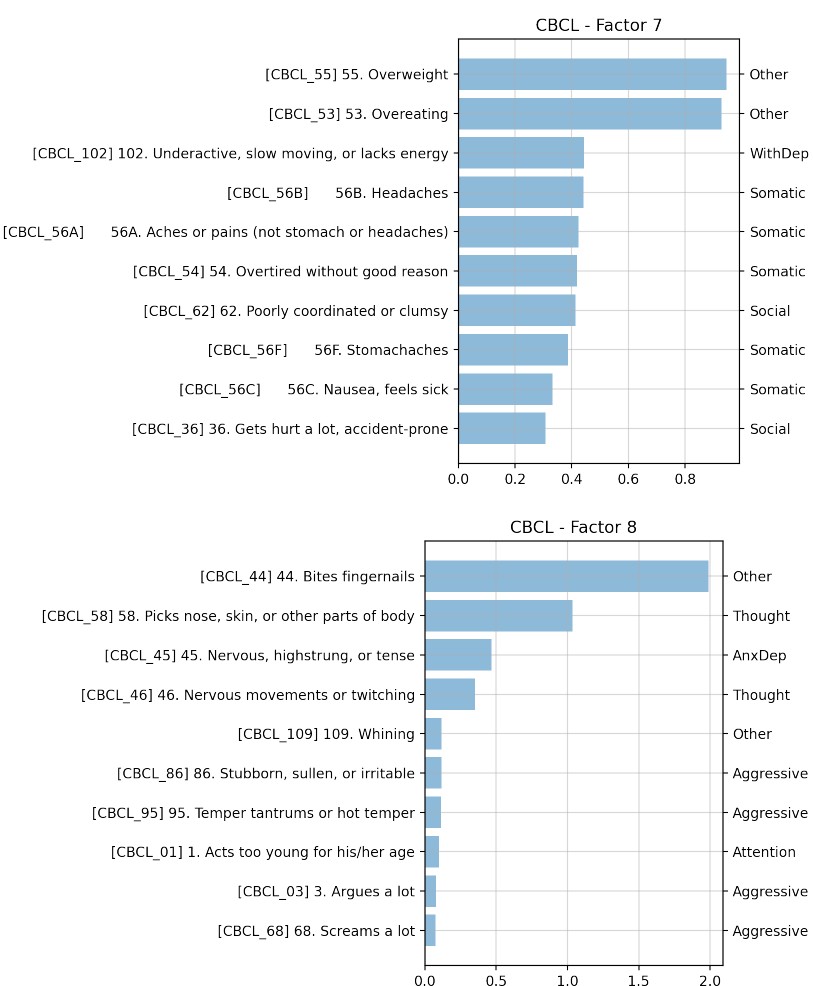

Figure 9: Top 10 questions ranked by *Q* in *CBCL* using *Q* obtained from ICQF (Factor 7-8).

| Factor | Theme |
|---|---|
| CBCL Factor-1 | irritability and oppositionality |
| CBCL Factor-2 | anxiety |
| CBCL Factor-3 | inattention and hyperactivity |
| CBCL Factor-4 | cognitive problems, disociality and callousness |
| CBCL Factor-5 | cognitive + fine motor problems |
| CBCL Factor-6 | body-focused repetitive behaviors |
| CBCL Factor-7 | somatic problems |
| CBCL Factor-8 | body-focused repetitive behaviors |

Table 4: Themes assigned to each CBCL factor by domain experts, summarizing the corresponding latent factors based on their components.

We also carried out subjective evaluations of the factor loadings produced by our method and factor analysis. Our clinical collaborators observed that:

- ICQF groups questions that are strongly related to cognitive impairments (which increase risk for psychopathology) into a separate factor (F5) whereas these are distributed across multiple factors in FA

- ICQF recovers the well-known dual comorbidly of depressive features with both anxiety symptoms (F2) and externalizing symptoms (F4) , whereas these dual links are missed by FA which puts most WithDep items into one factor with little other loadings (F4)

- ICQF groups distinctive body-focused repetitive behaviors into a distinct factor (F8) whereas these are not well differentiated by FA

- ICQF concentrates questions related to proactive aggression and antisociality into a single factor (F1) with captures the closely related feature of impulsivity. In contrast, FA distributes questions related to aggression into two factors (F1 and F13) which are weakly differentiated with regard to their loadings for other questions.

