# OpenReview forum: "Interpretable factorization of clinical questionnaires to identify latent factors of psychopathology"
_TMLR — Accepted by TMLR_

### Review · Reviewer_dnAi · 2026-03-19

**Summary Of Contributions:**

This paper proposed a non-negative matrix factorization method tailored for psychiatric questionnaire data, named ICQF. It constrains factor scores to [0,1], keeps loadings in the questionnaire's answer range, handles missing data natively, and models known confounders (age, sex) as additive components. ICQF is evaluated on the CBCL questionnaire across two large independent datasets (HBN and ABCD), outperforming standard factor analysis and $l_1$-NMF on diagnostic prediction and factor and loading stability.

**Audience:**

Yes

**Audience Explanation:**

The efforts to (1) impose constraints on both reconstructed matrix and factor and loading weights, and (2) incoporate confounder seem to be specifically designed for the domain of psychiatric questionnaire data, which I appreciate.

**Claims And Evidence:**

No

**Claims Explanation:**

**Concerns:**

1. More accurate reframing of the contribution. While the two efforts mentioned above are non-standard in the NMF literature, the rest of contributions highlighted in the paper are overly claimed. In particular, ADMM for NMF, sparsity penalty on $W$ and $Q$, missing data handling via a mask, blockwise cross-validation for rank selection, none of these are novel in isolation. The paper cites them as prior work and largely build the proposed method upon them.

   Also, while the abstract claims "Specifically, we show that ICQF improves interpretability, as defined by domain experts", it turns out the experts are collaborators of the authors, which substantially undermines the significance of the work.

   With that said, I encourage the authors to more accurately present their own contributions.

**Requested Changes:**

Besides the concerns above, please also address the following:

2. Clarification questions:
      - While it is stated "Factor loadings are bounded within the same value range as the original questionnaire responses", I did not find such restriction in the optimization objective (ICQF) and the following optimization procedures. Can authors clarify this point?
      - Can authors elaborate "The first is a linear combination of factors being a good approximation, which is the case for questionnaires." in Page 6?
      - Could you please elaborate the details of producing $W^{held-out}$?


3. Questions regarding experiments.

   - It may be unfair to use $\beta=0.1$ for both ICQF and $l_1$-NMF, as their penalty terms are different. Why not use cross validation to automatically select it as implemented in other experiments?

   - Figure 2, for a fair comparison, I suggest also compare factor loadings of ICQF with that of $l_1$-NMF, as $l_1$-NMF offer interpretation advantages such as non-negative loadings and sparsity as well.

   - Regarding experiment 2, the setting is not fully clear to me. Matrix decomposition can be done with both training and test data, thereby affecting the prediction power. If I understand correctly, in this paper, matrix decomposition is performed on training data solely, which may influence both the training of logistic models and the value of $W^{held-out}$.


4. Typo:

   - "capture capture" in section 3.1.
   - $C \in [0,1]$ in Page 3, $C$ should be a matrix.
   - Page 7, "... five popular detection algorithms", only four methods are mentioned.

---

> ### Author Response · Authors · 2026-05-04
> **Responding concerns and questions from Reviewer dnAi**
>
> We appreciate the reviewer's careful consideration of our paper, and their comments, suggestions, and questions, which will help us improve the clarity of the manuscript. We will begin by addressing the two main concerns, and answer other questions after that. All the typos identified have been fixed.
>
> **Concern: More accurate reframing of the contribution**
>
> We agree with the reviewer that the original wording on the novelty of the contribution was not clear. To be precise, we believe that our contribution lies in the unified ICQF framework, the theoretical results validating its fitness for purpose, and its empirical evaluation. All of ADMM for NMF, sparsity regularization, masking for missing entries, and blockwise cross-validation are established ideas, and our work builds on them to reach our goal. We will revise the manuscript to clarify this and, specifically, emphasize the following technical contributions:
>
> - we derive a sufficient condition on the ADMM penalty parameter, namely $\rho = \sqrt{2}$, under which the ADMM iterates are guaranteed to converge to a KKT point for our constrained formulation;
> - using realistic synthetic data, and stated regularity conditions, we show that the algorithm converges to the ground truth under this model;
> - we show that these theoretical results extend naturally to the inclusion of simple box constraints, which improve interpretability while remaining compatible with the framework.
>
> **Concern: the experts verifying the interpretability of ICQF are collaborators of the authors**
>
> We agree with the reviewer that the wording in the abstract should have emphasized that the domain experts are our clinical collaborators (and co-authors), beyond the current mention in the discussion.  We will revise the manuscript accordingly. The development of our method was driven by this collaboration, from the initial desiderata to the quantitative experiments and visualizations. By including their qualitative assessments, we aimed at providing a perspective that the models learned were reasonable and indeed satisfied those desiderata. This was not meant as the primary evaluation, of course; that would be the quantitative evaluation carried out in both synthetic and real data.
>
> **Question: I did not find such restriction of factor loadings in the optimization objective (ICQF) ...**
>
> Thank you for pointing this out, as this should indeed have been described more clearly. The boundedness of the factor loadings is enforced through the constraint set in the main ICQF formulation, namely
> $$\mathcal{Q} = \{ Q \vert \ 0 \leq Q_{ij} \}$$
> where $Q_{ij}$ represents the upper bound associated with the questionnaire response range. In other words, the optimization is carried out under an explicit box constraint, so the factor loadings are restricted to lie within the same admissible range as the corresponding questionnaire responses.
>
> We regret the confusion caused by presenting this constraint in its most general and flexible form. Our intention was to allow the formulation to accommodate even subject-specific and item-specific response ranges, which could be represented through $Q_{ij}$. However, in most practical questionnaire settings, the response range is only item-specific rather than subject-specific. Therefore, it is indeed more natural and easier to interpret this constraint using MATLAB-style notation as $Q_{*j}$, indicating that the upper bound depends only on question item $j$. We will revise the text to make this point explicit.
>
> **Question: Can authors elaborate "The first is a linear combination of factors being a good approximation, which is the case for questionnaires.**
>
> Thank you for pointing this out, we apologize for the unclear description. What we intended to convey is that the first assumption is that
> $$\mathbf{M} = \mathbf{W}^* \left( \mathbf{Q}^* \right)^{\top}$$
> provides a good approximation to the observed data matrix $\mathbf{M}$, that is, the data can be well represented by a linear combination of latent factors. For questionnaire data, this is often a reasonable modeling assumption, since the observed responses are typically viewed as arising from a small number of underlying latent constructs, and the questionnaires were designed and validated with that in mind. By ``second'', we were referring to the second assumption, namely that the latent dimension $k^*$ can be accurately estimated. We will revise the text to make the distinction between these two assumptions explicit.

---

> > ### Author Response · Authors · 2026-05-04
> > **Responding concerns and questions from Reviewer dnAi (Part II)**
> >
> > **Question: Could you please elaborate the details of producing $W^{\text{held-out}}$?**
> >
> > We appreciate your pointing this out, as we now see how this could be confusing. For the synthetic data, the held-out set $W^{\text{held out}}$ can be generated at the data creation stage, following the same distribution rule described in Equation (18). After generating the full dataset, we simply split $W$ into training and held-out portions according to the 80/20 rule. That said, for the synthetic example in Section 4.1, our main focus is on estimation of the latent dimension $k^{*}$, rather than inference of $W$ on a held-out dataset.
> >
> > Therefore, in that section we effectively treat the full $W$ with 200 samples as the training set, and for each algorithm we estimate $k$ and examine how well the resulting estimate aligns with the ground-truth $k^{*}$. We will revise the manuscript to better explain how $W^{\text{held-out}}$ is generated, and to clarify the role of the held-out set in Section 4.1.
> >
> > **Question: Why not use cross validation to automatically select it as implemented in other experiments?**
> >
> > We agree with the reviewer that it would be unfair if done that way, and hence apologize for the lack of clarity that led to that misunderstanding. It likely comes from the wording in the first paragraph of Section 4. Our statement that the methods use the ''same sparsity regularization strength ($\beta = 1e-1$)'' was intended only for the settings where BCV is not involved. In those cases, the same fixed value is used for both L1-NMF and ICQF as a common experimental setting. When parameter tuning is performed, we do use BCV to select the regularization parameter automatically; this is done elsewhere in the paper, for example in Appendix F. We will revise the manuscript to clarify that the fixed ($\beta = 1e-1$) setting applies only to experiments where BCV is not used, while BCV-based tuning is adopted in other experiments.
> >
> > **Question: Figure 2, for a fair comparison, I suggest also compare factor loadings of ICQF with that of $\ell_1$-NMF**
> >
> > Thank you for this suggestion. When preparing the initial version of the paper, we debated whether to include factor loadings from $\ell_1$-NMF, in addition to ICQF and factor analysis. We opted not to, since the comparison with factor analysis might be the most relevant for users engaged in a clinical application. We do agree that the comparison would be enriched by including it, and will therefore add the corresponding factor loadings to the revised manuscript.
> >
> > Given your interest, we will also include them here for your consideration. We used BCV with $\ell_1$-NMF and obtained the estimated optimal parameters $\beta=0.0$ and $k=11$. It is noteworthy that BCV selected $\beta=0.0$, meaning that no additional $\ell_1$-based sparsity regularization is supported by this data-driven procedure in this setting. In addition, factors with relatively large magnitude in $Q$ may dominate the factor loading heat map visualization. Although one could manually tune $k$ downward or apply post hoc rescaling to obtain visually denser or more balanced loading patterns, such adjustments would no longer be purely data-driven in the same way that the BCV-selected solution is. In particular, matrix factorization is scale-indeterminate: rescaling a column of $Q$ necessarily induces an inverse rescaling of the corresponding column of $W$ while leaving the product unchanged. Therefore, while such post hoc normalization may improve visual comparability, it also changes the scale of the latent scores in $W$, which may affect downstream applications where $W$ itself is interpreted or used as input. This need for post-hoc transformation is also present in factor analysis, and was one of the motivations for our development of ICQF.
> >
> > More broadly, this result suggests that BCV, which is designed around predictive reconstruction performance, is not necessarily aligned with the interpretability objective for $\ell_1$-NMF in this application. We will clarify this point in the revised manuscript.

---

> > > ### Author Response · Authors · 2026-05-04
> > > **Responding concerns and questions from Reviewer dnAi (Part III)**
> > >
> > > **Question: Regarding experiment 2, the setting is not fully clear to me...**
> > >
> > > Thank you for raising this issue. This is indeed an important experimental detail, and therefore needs to be explained more clearly. We will revise the manuscript to make this protocol explicit, and will provide a succinct explanation here for your consideration.
> > >
> > > In our setting, a matrix decomposition that jointly learns both $W$ and $Q$ is always carried out on the training data alone. Here, $W$ represents the subject-specific latent representation, while $Q$ represents the factor loadings shared across subjects. $Q$ is used in inference of latent representations for test subjects. Therefore, it would not be appropriate to use the test data when learning $Q$, since that would introduce information leakage into the prediction pipeline.
> > >
> > > In more detail, when inferring latent representations for new held-out subjects, we assume that $Q$ is fixed, after being learned from the training set. We then optimize the ICQF objective with $Q$ fixed and update only $W$ (as shown in the Figure Appendix E) for subjects in the validation or held-out datasets. We denote the latent representations learned as $W^{\text{validate}}$ and $W^{\text{held-out}}$ respectively. Under this setup, the size of the training set can affect how stably and accurately $Q$ is estimated. If $Q$ is estimated less accurately due to limited training data, then the inferred $W^{\text{validate}}$ and $W^{\text{held-out}}$ may also be less accurate, which can in turn affect the downstream prediction performance. This is the point of Experiment 2 where, by varying the amount of available training data, while keeping a fixed test set,  we can evaluate how well the different methods can still learn to recover a latent representation carrying diagnostic information., Specifically, we keep 20\% of the full dataset as the test set throughout. Then, for each training proportion (80\%, 60\%, 40\%, and 20\%), we repeat the full procedure -- learning the decomposition on the corresponding training subset, inferring $W^{\text{held-out}}$ for the fixed test set with $Q$ held fixed, training the prediction model using the training/validation split, and then evaluating diagnostic prediction performance on the same held-out test set -- to obtain the reported prediction trends.

---

> > > > ### Comment · Reviewer_dnAi · 2026-05-05
> > > >
> > > > I thank the authors for their detailed responses. Most of my concerns have been addressed. I have a few follow-up comments and questions:
> > > >
> > > > - Constraint in ICQF. The authors explain that $Q_{ij}$ represents a range constraint. However, this notation is potentially confusing, as $Q$ is also the loading matrix to be optimized. I would recommend using a different symbol for the constraint to avoid ambiguity.
> > > >
> > > > - Hyperparameter tuning. In the current manuscript, it appears that BCV is used solely for selecting $k$. However, the authors indicate that BCV is also used to tune $\beta$. This should be clarified in the revision.
> > > >
> > > > - Comparison with NMF. It is interesting that cross-validation selects $\beta=0$. Otherwise, one might expect the resulting $Q$ to resemble that obtained from ICQF in terms of interpretation. It would be helpful to clarify what specific advantages ICQF provides over NMF for non-zero penalties.
> > > >
> > > > - Co-authors as domain experts. My point is, without independent real-world validation, it is difficult to assess the strength of the claims made in the paper. While the proposed method may offer appealing interpretability in certain settings, its broader impact and practical significance remain somewhat unclear.

---

> > > > > ### Author Response · Authors · 2026-05-07
> > > > > **Responding follow-up comments and questions from Reviewer dnAi**
> > > > >
> > > > > We thank the reviewer again for the careful follow-up comments and going through our previous responses in details. We are glad to see that most of the reviewer's concerns have been addressed, and we respond below to the remaining follow-up comments and questions.
> > > > >
> > > > > **Comment: I would recommend using a different symbol for the constraint to avoid ambiguity.**
> > > > >
> > > > > We agree that the current notation is potentially confusing, since $Q$ is used for the loading matrix and for the range constraint, and thank the reviewer for pointing this out. We will revise the manuscript to use a different symbol for the bounded constraints, so that the optimization variable and the constraint notation are clearly distinguished.
> > > > >
> > > > > **Comment: Hyperparameter tuning. In the current manuscript, it appears that BCV is used solely for selecting $k$. However, the authors indicate that BCV is also used to tune $\beta$. This should be clarified in the revision.**
> > > > >
> > > > > We agree that this should be clarified. In the revision, we will make explicit that BCV is used not only for selecting $k$, but also for tuning the sparsity regularization parameter $\beta$ in the experiments where hyperparameter tuning is performed. Our aim was to make it possible to have most parameters be determined from data, if feasible.
> > > > >
> > > > > **Question: It would be helpful to clarify what specific advantages ICQF provides over NMF for non-zero penalties.**
> > > > >
> > > > > Thank you for raising out this helpful point. The key distinction is that ICQF is not defined only by sparsity regularization, but by a different constrained optimization problem. In particular, ICQF imposes explicit range constraints on the factor loadings throughout estimation, whereas standard NMF and $\ell_1$-NMF do not. Therefore, even when the selected sparsity penalty is zero or small, the feasible set of ICQF remains strictly more structured, and incorporates application-specific prior knowledge about the admissible range of questionnaire responses. Hence, the distinction from NMF is not just the presence or absence of the $\ell_1$ penalty, but the geometry of the feasible set itself. In other words, the difference in $\beta$ does not imply that sparsity is uniformly useful or useless across methods.
> > > > >
> > > > > Specifically, BCV selects hyperparameters based on reconstruction/prediction performance alone, rather than interpretability. Therefore, the selected $\beta$ reflects the trade-off for that criterion within each method's own optimization problem. In $\ell_1$-NMF, the loading matrix is constrained mainly by non-negativity, so adding sparsity may not improve the BCV criterion and can therefore lead to the selection of $\beta=0$. In ICQF, the loading matrix is additionally restricted by the explicit range constraints, which already enforce scale-consistent and application-aligned solutions. Within this more structured feasible set, a nonzero $\beta$ can still be preferred because it promotes a simpler and more localized factor structure without leaving the admissible questionnaire-response range. The practical advantage of ICQF is therefore that interprertability-relevant structure is built directly into the factorization itself, rather than relying on sparsity alone or on post hoc rescaling of the learned loadings. We will revise the manuscript accordingly and include this clarification, alongside the added NMF results.
> > > > >
> > > > > **Comment: Co-authors as domain experts. My point is, without independent real-world validation, it is difficult to assess the strength of the claims made in the paper. While the proposed method may offer appealing interpretability in certain settings, its broader impact and practical significance remain somewhat unclear.**
> > > > >
> > > > > We agree with the reviewer that it would be ideal to have independent validation of the interpretability of our factorization versus exploratory factor analysis, in this and, ideally, other questionnaires. However, we think such a rigorous study is beyond the scope of the current work, as the primary contribution is methodological (even if the motivation comes from psychiatry). This said, we are  already deploying the method in two separate psychiatric research collaborations, where our collaborators think the method yields interesting -- and clinically plausible -- results. In each case, we are comparing the ICQF and exploratory factor analysis solutions for each questionnaire in a set covering several different aspects of psychopathology. In tandem, we will consider the suitability of the resulting factor scores for downstream uses such as prognostic prediction. We think that the diversity of questionnaires in either project will provide a more representative setting in which to assess the interpretability of the factor loading patterns identified, as well as other measures of validity.

---

### Review · Reviewer_a2UZ · 2026-03-24

**Summary Of Contributions:**

The goal of the paper is

1. handle missing data;
2. handle confounders;
3. And to increase the interpretability of the factors

in the context of factor analysis. to this end, they introduce an optimization procedure where they try to minimize reconstruction loss of some input matrix $M$, adding 1. a mask to handle missing entries and 2. a loading on confounders that are not accounted for in $M$, but are provided by prior knowledge. The resulting optimization procedure comes with theoretical guarantees.

**Audience:**

No

**Audience Explanation:**

I replied no, with one caveat: I am no expert at all in clinical questionnaires, so I won’t argue about that, leaving it to more expert reviewers.
However, __from the perspective of latent variable model inference__ (factor analysis being one instance of the problem), __I have several concerns__ about the optimization procedure and the authors claim. In reference to section 3.1

1. If I understand correctly, the confounders $C$ come from prior knowledge. I wonder what happens if we miss some confounder in $C$. This should be discussed and empirically verified
2. Authors say that they constrain "entries of Q being in the same value range as question answers, so loadings are interpretable": why this choice of constraining the loading matrix to the value range of questions would enhance interpretability? Scaling is one intrinsic indeterminacy of factor analysis. How do they know that imposing this prior is the right choice to select the "right" scaling values? It seems like an arbitrary choice of choosing one solution among the many, indistinguishable ones
3. A similar comments goes on the regularization parameter $\gamma$, which the authors claim mitigates scale ambiguity. I don’t see it: regularization is one way to choose a solution for an ill-posed problem. Again, in the context of FA, we know that this ambiguity is somewhat irreducible in the absence of restricting assumptions. Why do you think that the imposed regularization leads to a sensible choice of values of scale of $Q$?
4. Finally, the authors add sparsity constraints on the loading matrix, without a clear discussion on this. Can they please argue?

All in all, given these concerns, I don’t see a clear contribution to the factor analysis inference problem — specifically, about mitigating the intrinsic ambiguities.

Moreover, __I don't see the pertinence__ of the focus on clinical questionnaire in psychopathology __for a TMLR submission.__

**Claims And Evidence:**

Yes

**Claims Explanation:**

The empirical evidence supports the theoretical claim of the overperformance of their optimization procedure versus standard methodologies.

**Requested Changes:**

1. empirical on the role of confounders in $C$; what if some confounders is omitted?
2. Discussion on why their optimization procedure is the right one to solve scaling ambiguities

In my opinion, this would strengthen the submission. but concerns remain whether TMLR is the right venue for the submission

---

> ### Author Response · Authors · 2026-05-04
> **Responding questions from Reviewer a2UZ**
>
> We appreciate your careful reading of our paper, and the questions that have helped us identify points that need clarification or further elaboration. We will endeavor to answer all your questions below, starting with your most general comment.
>
> **Comment: I don't see the pertinence of the focus on clinical questionnaire in psychopathology for a TMLR submission**
>
> Questionnaire data are present in many domains beyond clinical psychiatry research (e.g. surveys, filling a structured information grid from unstructured text such as interviews or notes). Domain experts typically have a battery of questions about observable phenomena, and wish to find if a few latent factors would explain the pattern of answers. We are confident that there are many applications involving questionnaires, and hence the pertinence really boils down to whether those applications warrant a new method. This is a latent variable modeling problem, in general, and one where any linear factorization could be deployed. Given the need to address indeterminacy, as the reviewer points out, and also incur in post-hoc work to make the solutions interpretable, we believe that there is room for a method that incorporates interpretability desiderata directly as factorization constraints and regularization. We think this would be intrinsically useful in any application domain with questionnaire data. In the clinical psychiatry domain, it also appears to be more efficient in recovering ground truth in realistic synthetic data, or preserving diagnostic information in small sample sizes. We developed the method because we needed it for our own application work, otherwise we would have used an existing one.
>
> **Question: I wonder what happens if we miss some confounder in $C$**
>
> We thank the reviewer for bringing this point up, as it definitely requires clarification, which we will add in the revised manuscript. The matrix $C$ is meant to contain known observed covariates (e.g., age, sex), as stated in Sec. 3.1. In questionnaire datasets, these standard demographic confounders are generally available, so our experiments assume $C$ is observed. The intent of having this in the model is to capture baseline patterns of answers that are driven by those covariates, so that the other factors model variation beyond that. We included this because Factor Analysis does not: typically, either covariates are ignored outright, or the samples are separated by covariate (e.g. sex or age range). This leads to potentially different models being learned in each subsample, from less data, and with an additional need for post-facto harmonization of common factors. In our scenario, what matters is the ability to model the influence of known covariates; this is not like causal inference, for instance, where there is often an assumption of no unobserved confounders.
>
> If we misunderstood the reviewer, and the concern is instead that the covariates might be missing for some participants, we believe that can be addressed within our framework as well. If some entries of observed $C$ are missing, two extensions are possible: jointly estimate/impute them within the factorization framework, or use simple preprocessing imputation before factorization.

---

> ### Author Response · Authors · 2026-05-04
> **Responding questions from Reviewer a2UZ (Part II)**
>
> **Question: why this choice of constraining the loading matrix to the value range of questions would enhance interpretability? How do they know that imposing this prior is the right choice to select the "right" scaling values?**
>
> We agree with the reviewer that scaling is one intrinsic indeterminacy of factor analysis. This is why exploratory factor analysis generally requires steps of rescaling and rotation: they eliminate differences in range for factor scores, ensure those factors and corresponding loadings are sign-aligned, and optionally rotate so that items are associated with as few factors as possible. Our argument is that all of these are usually done for the sake of interpretability, and therefore it is reasonable to consider embedding them as part of the method, rather than making it a post hoc step. Our motivation originates in principle -- what do our collaborators tell us they would like to see in a model -- and convenience versus exploratory factor analysis.
>
> To directly answer the reviewer's question as to why constraining the loading matrix would favour interpretability, this follows from the joint parameterization: factor scores $W \in [0, 1]$ represent degree of factor presence; $M \in [\min(M), \max(M)]$ means reconstruction stays within the questionnaire response range; $Q$ is bound by the item response range.
>
> The constraint not posed in isolation; it works with the other constraints to promote the intended semantics of the model. It aims to favour interpretability in two ways. First, a loading does not take a value in an arbitrary range, with a positive or negative sign, as could happen in factor analysis; it is, instead, something directly legible in the scale the questions are written in, as a ''prototypical'' answer presentation pattern for the factor.
>
> Second, by keeping loadings positive it encourages sparsity, since they can no longer be traded-off between two factors with negative and positive scores. They can only be contributed by each factor additively, and with the total also constrained.
>
> Finally, we agree with the reviewer that, absent any further evidence, it would be hard to know if imposing this prior was the "right" choice. However, we believe it is possible to quantitatively test if the prior we impose is "right" or, at any rate, closer to the underlying truth than those imposed by other methods. The synthetic data experiment described in Section 4.1 tests recovery of the ground truth. The experiment described in Section 4.2.4, and illustrated in Figure 3, tests whether the ICQF is more efficient at recovering latent variables carrying diagnostic information, as the amount of training data decreases, for a fixed test set.
>
> **Question: A similar comments goes on the regularization parameter $\gamma$...**
>
> We agree with the reviewer that regularization does not remove scale ambiguity in a strict identifiability sense. In our model, the parameter $\gamma$ has a narrower purpose: balancing sparsity regularization between $W$ and $Q$. Because $W$ and $Q$ have different dimensions and roles, an unbalanced penalty can shrink one side much more than the other. This can produce a solution where sparsity is expressed mostly in $W$ or mostly in Q, hurting interpretability. Therefore, $\gamma$ helps choose a more stable and sensible representative from the scale-equivalent solution family. The alternative would be to have separate penalties for $W$ and $Q$, which would be more cumbersome, and increase the cost of estimating the parameters from data.
>
> **Question: Finally, the authors add sparsity constraints on the loading matrix, without a clear discussion on this**
>
> We thank the reviewer for bringing up this point, as we agree that it needs to be elaborated in the revised manuscript. There are two primary reasons for using sparsity on the loading matrix $Q$. The first is to strive for a conservative solution, where we encourage factors to load on as few items as is advantageous to group together for reconstruction. This will tend to isolate core symptom groups that co-occur to some extent. The second is a consequence of the first, in that symptom groups may then be separated over more factors than they would in factor analysis. These are often subjectively easier to interpret, as follows. The loading magnitude reflects their relative importance/prevalence, as it is in a comparable scale for all of them. They may also be viewed a split of an existing "subscale" of symptoms in a questionnaire, which would typically be grouped together in factor analysis, as that is how questionnaires were designed. Identifying such data-driven reasonable splits, if and when they exist in certain clinical populations, was one of the reasons for developing the method.

---

### Review · Reviewer_n8Mg · 2026-04-21

**Summary Of Contributions:**

**Summary**

This paper proposes Interpretable Constrained Questionnaire Factorization (ICQF), a non-negative matrix factorization framework tailored for clinical questionnaire data. The key idea is to incorporate domain-motivated constraints (boundedness, non-negativity, sparsity, and confound modeling) into the factorization process to improve interpretability and stability.

The authors formulate a constrained optimization problem and solve it using an ADMM-based algorithm with convergence guarantees. They further propose a blockwise cross-validation (BCV) procedure to estimate the latent dimensionality. Empirical evaluation on synthetic data and real clinical datasets (HBN and ABCD) demonstrates improved interpretability, preservation of diagnostic information, and robustness under small sample sizes compared to baseline methods.

**Strengths**

The paper is well-motivated by a real-world clinical need: interpretable latent factor discovery from questionnaire data.

The incorporation of domain-specific constraints is intuitive and practically meaningful.

The experimental evaluation is comprehensive, including synthetic validation and multiple real-world datasets.

The method demonstrates consistent improvements in predictive performance and stability across settings.

**Weaknesses and Questions**

A key limitation of the paper is the lack of a clearly stated data-generating mechanism. The model assumes a factorization of the form $M \approx W Q^T$, possibly augmented with confounders, but it remains unclear whether this formulation reflects a realistic generative process for questionnaire data. In particular, it is not specified whether questionnaire responses are assumed to be linear combinations of latent factors, nor how confounders are incorporated into the generative process.

In addition, the method relies heavily on constraints such as $W \in [0,1]$, bounded $Q$, and bounded reconstruction $Z$. While these constraints are motivated by interpretability considerations, the paper does not provide sufficient theoretical justification for why they are appropriate or necessary.

Finally, the novelty of the theoretical analysis appears limited. Proposition 3.2 shows that incorrect specification of the latent dimension leads to increased reconstruction error, but this is a well-known phenomenon in matrix factorization and does not seem specific to the proposed method.

**Audience:**

Yes

**Audience Explanation:**

Yes. I expect this paper to be of interest to at least part of the TMLR audience, especially researchers working on matrix factorization, interpretable representation learning, computational psychiatry, and machine learning methods for clinical or behavioral data. The problem setting is concrete and practically relevant, and the paper connects methodological ideas from constrained factorization with an application area where interpretability, missing data, and confounding are central concerns.

**Claims And Evidence:**

Yes

**Claims Explanation:**

Mostly yes. The paper provides multiple forms of evidence that are relevant to its main claims. On the empirical side, the method is evaluated on both synthetic data and real CBCL data from two independent cohorts, and the reported results show improved diagnostic-information preservation and more stable factor loadings relative to the baselines in several settings. The qualitative comparison with factor analysis and the appendix-based expert interpretation also support the claim that the learned factors may be easier to interpret.

Some of the stronger theoretical positioning seems less well substantiated than the empirical claims. Proposition 3.1 is useful in establishing optimization behavior, but Proposition 3.2 appears more indirect and its connection to what is specific to ICQF is not entirely clear.

**Requested Changes:**

1. The paper should clearly specify the assumed data-generating mechanism underlying the proposed factorization.

2. The authors should provide stronger justification for the imposed constraints on $W$, $Q$, and the reconstruction.

3. The novelty of the theoretical analysis should be clarified. In particular, Proposition 3.2 appears to reflect a well-known property of matrix factorization rather than a result specific to ICQF. The authors should either strengthen the theoretical contribution or clearly position it as a limitation.

---

> ### Author Response · Authors · 2026-05-04
> **Responding weaknesses by Reviewer n8MG**
>
> We appreciate the reviewer's careful consideration of our paper, and especially bringing up the point below. It made us realize we should elaborate on the underlying assumptions about the generative process that are usually made when modeling questionnaire data, as these may be less familiar to a machine learning audience. We will revise the paper to address this, with the content provided in this answer, and answer all other questions after this.
>
> **Weakness: A key limitation of the paper is the lack of a clearly stated data-generating mechanism...**
>
> When exploratory factor analysis (EFA) is used on questionnaire data, the model assumes a factorization of that form, with normally distributed, uncorrelated latent factors, fixed loadings, and item specific noise. Accounting for items being discrete, ordinal answers complicates things to some extent, but the framing is still a linear factorization. Each factor is independently responsible for increasing/decreasing the presence of an item and, given that loadings have real values, the influences of different factors can trade-off against each other. In our view, this framing is more appropriately understood as a modeling assumption for handling latent factors that may manifest across many items in different combinations, across study participants, rather than as a full-fledged probabilistic generative model.
>
> Our framing is also a linear factorization, with the modifications aiming at addressing some of the aspects of EFA that complicate interpretability. If we were to take a generative perspective, the starting point would be to say that each factor can be present in a participant to a varying degree (from 0, absent, to 1, fully present). The extent to which multiple factors may be present at once is determined from data, rather than specified; the implicit prior is that only a few will be present at once for each participant, unless the evidence is overwhelming. Each factor can potentially manifest across all items, and the influence of multiple factors combines additively. From that perspective, the loadings associated with confounds can be viewed as baseline rates of presence of an item, for the subgroup sharing that confound. Because of the nonnegativity constraints, multiple factors cannot have mutually-cancelling influences on an item; they can only contribute additively, with loadings in the same range as the items, and with the total also constrained.
>
> This promotes interpretability in at least two ways. First, it encourages factors to specialize on subsets of items characteristic to them, rather than overfit to co-occurrences by chance, which is particularly important for small samples. Second, it makes the loadings legible as an answer pattern, which could be read as a "prototypical" item presentation pattern for the factor. So, in conclusion, the formulation and constraints were chosen primarily to make the model more identifiable and promote interpretability, rather than to instantiate a more complex generative model.

---

> ### Author Response · Authors · 2026-05-04
> **Responding weaknesses by Reviewer n8MG (Part II)**
>
> **Weakness: the paper does not provide sufficient theoretical justification for why the constraints are appropriate or necessary...**
>
> We hope that, in providing a generative perspective, the previous answer also provides a view on the appropriateness of the modelling assumptions from a clinical perspective. We are more than willing to elaborate further if the reviewer deems it necessary.
>
> Beyond interpretability, there are additional advantages to using these modelling assumptions. First, keeping factor scores $W$ bounded and in the same range provides a "well-behaved" representation for downstream applications. Examples of these would be diagnostic or outcome prediction models, or models relating the information in questionnaires to brain imaging or genetics measures. Second, they help stabilize the ADMM updates. In a nonconvex, unconstrained factorization, there can be extreme intermediate rescaling of $W$ and $Q$ with nearly unchanged reconstruction, which can make the subproblems poorly scaled and the primal/dual residuals less balanced. This can increase the sensitivity of the algorithm to the choice of $\rho$, as well as to initialization and stopping behavior. The bounded constraints enforce the iterates back into a compact feasible region at every step, limiting extreme values and keep optimization more stable in practice. As a result, the penalty parameter $\rho$ does not need to be set overly large in order to control unstable iterates or enforce feasibility. This is practically useful because an overly large $\rho$ can cause the augmented penalty term to dominate the updates, making the iterates less adaptive and potentially leading to poorer stationary solutions. Boundedness helps reduce scale ambiguity between $W$ and $Q$ and discourages degenerate rescalings, while non-negativity enforces additive latent contributions (a form of "parts based representation") that are easier to interpret clinically. We agree with the reviewer that these constraints alone do not guarantee uniqueness or full identifiability, and we will clarify this point in the revision.
>
> **Weakness: Proposition 3.2 shows that incorrect specification of the latent dimension leads to increased reconstruction error, but this is a well-known phenomenon in matrix factorization and does not seem specific to the proposed method...**
>
> We agree with the reviewer that the general phenomenon is well known in matrix factorization, and applies broadly. Our intention was rather to clarify that this issue remains relevant in the ICQF setting, despite additional boundedness and nonnegative constraints imposed on the factorizations.  Proposition 3.2 is meant to address this constrained setting in particular.
>
> This specific issue matters for our application because, when $k$ is misspecified, the inferred latent representation might depart substantially from the ground-truth and from the clinically meaningful structure we seek to recover. For instance, when $k > k^*$, a coherent latent factor may be split into several more specific components. While these factors would still be present in the data, rather than the product of overfitting, splitting a coherent symptom presentation would hinder clinical research. This is also why data-driven selection of $k$ such as BCV is important, instead of choosing $k$ ad hoc based on a preferred qualitative structure, or relying on external methods (e.g. BIC). The goal of the synthetic data experiments was to determine whether the use of BCV led to better estimates of $k$ than alternative methods used with FA or NMF and, if so, whether the ground truth latent structure could be recovered.
>
> More broadly, Proposition 3.1 provides the optimization guarantee, and Proposition 3.2 provides model-selection motivation: together with the known star-convexity result when $k = k^*$, they support the importance of selecting $k$ correctly in order to obtain a meaningful ICQF solution. If so, the combination of propositions 3.1 and 3.2 should increase our confidence that a factorization with our assumptions will converge to a reasonable solution rather than a degenerate local optimum. This said, we agree that this proposition should be viewed mainly as a supporting result for model selection motivation, rather than as a central theoretical novelty claim. In the revision, we will shorten this discussion and use the space to clarify modeling assumptions and the role of the imposed constraints instead.

---

### Decision · Action_Editor_3qqZ · 2026-06-15

**Recommendation:** Accept as is

**Audience:**

Yes

**Audience Explanation:**

Interpretability and tabular data are of broad interest across many domains.

**Claims And Evidence:**

Yes

**Claims Explanation:**

The authors propose a method to find latent factors in clinical questionnaires. They provide strong empirical evidence that it works. Two reviewers agree with the authors. Reviewer a2UZ notes that interpretability lacks a theoretical definition. This is a fair point. However, I side with the positive reviewers. Interpretability is not the same as identifiability. It is currently an empirical matter.

---

> ### Author Response · Authors · 2026-06-16
> **Author response to final decision**
>
> Dear Editor and Reviewers,
>
> We would like to sincerely thank you for your time and thoughtful evaluation of our manuscript. We greatly appreciate the reviewers' valuable comments and suggestions throughout the review process. The questions and feedback have helped improve the clarity and scientific reliability of the manuscript. As committed during the review discussion, we will incorporate the updates into the final version of the manuscript before publication.
>
> Thank you again for your constructive feedback and for accepting our work for publication.
>
> Sincerely,
> The Authors